

# Ground-state energy and excitation spectrum of the Lieb-Liniger model: accurate analytical results and conjectures about the exact solution

**Guillaume Lang[1,2], Frank Hekking[1,2] and Anna Minguzzi[1,2]**

**1** Université Grenoble Alpes, LPMMC, F-38000 Grenoble, France
**2** CNRS, LPMMC, F-38000 Grenoble, France

## Abstract

We study the ground-state properties and excitation spectrum of the Lieb-Liniger model, i.e. the one-dimensional Bose gas with repulsive contact interactions. We solve the Bethe-Ansatz equations in the thermodynamic limit by using an analytic method based on a series expansion on orthogonal polynomials developed in [1] and push the expansion to an unprecedented order. By a careful analysis of the mathematical structure of the series expansion, we make a conjecture for the analytic exact result at zero temperature and show that the partially resummed expressions thereby obtained compete with accurate numerical calculations. This allows us to evaluate the density of quasi-momenta, the ground-state energy, the local two-body correlation function and Tan's contact. Then, we study the two branches of the excitation spectrum. Using a general analysis of their properties and symmetries, we obtain novel analytical expressions at arbitrary interaction strength which are found to be extremely accurate in a wide range of intermediate to strong interactions.



# 1 Introduction and motivation

The one-dimensional (1D) model of point-like bosons with repulsive contact interactions has been introduced in [2] as a generalization to finite interaction strengths of the Tonks-Girardeau (TG) gas of hard-core repulsive bosons [3], and is known in the literature as the Lieb-Liniger (LL) model or the $\delta$-Bose gas. Its exact ground state is encoded in a set of coupled equations obtained in [2] and [4] by coordinate Bethe Ansatz (BA). The LL model is one of the simplest quantum integrable models [5–8]. Its exact solution has helped understand various aspects of the many-body problem in one dimension, the most appealing features being the effective fermionization of bosons at large interaction strength [3,9] and the existence of two branches of excitations [4], one of them reminiscent of the Bogoliubov dispersion [2,10], the other linked with a quantum analog of classical solitons [11]. The equilibrium grand canonical description of the LL model has been developed by Yang and Yang who introduced the Thermodynamic Bethe Ansatz [12], thereby opening new investigation lines, such as finite-temperature thermodynamics [13–16] or quantum statistics of the model [17]. Later on, Haldane used the Lieb-Liniger model as a testbed for the universal description of low-energy properties of gapless 1D systems within the bosonization technique [18], known as the Tomonaga-Luttinger liquid (TL) framework [19–25].

    The calculation of the dynamical correlations of the LL model remained for a long time an open problem. The determination of its exact time-dependent density-density correlations and their Fourier transform, known as the dynamical structure factor, is easily tractable in the Tonks-Girardeau regime only [26]. Perturbation theory allows to tackle the strongly-interacting regime [27], and the TL predictions at arbitrary interaction strengths [18,28,29] are accurate only within a small low-energy range [30,31]. Finding the exact solution at arbitrary interaction strength actually required the development of fairly involved algebraic Bethe Ansatz (ABA) techniques relying on the quantum inverse scattering method [32]. The form factors were computed numerically using the ABACUS algorithm [33], first at zero [34,35], then at finite temperature [36], while mathematically-oriented works focus on analytic and algebraic general considerations [37–40]. All of them tend to validate a nonlinear extension

of the Tomonaga-Luttinger liquid theory developed in parallel [41].

The experimental progress in cooling and trapping ultracold atoms has led to a renewed interest for the LL model. Strong confinement in two transverse dimensions [42–44] and low enough temperatures have led to realizations of (quasi-)1D systems of bosons [45], some of them suitably described by the LL model [46]. The possibility to tune interaction strength in a controlled way [47] has allowed to probe any interaction regime from weak to strong [48,49], as well as attractive ones [50], yielding a strongly excited state called 'super-Tonks-Girardeau' (sTG) gas [51]. For a review of experimental studies of the properties of the LL model we refer to [52,53]. In particular, the momentum distribution has been measured [48], as well as local two- [54] and three-body correlations [55–57]. The phase diagram at finite temperature predicted in [58] has been explored in [59]. More recently, measurements of the dynamical structure factor have validated the ABA predictions in a regime where the standard Tomonaga-Luttinger liquid approach is not applicable [60,61]. In current experiments, it is also possible to investigate out-of-equilibrium dynamics [62,63]. From a theoretical point of view, the LL model allows to address a rich variety of topics, e.g. the effect of quantum quenches [64–69], few-body physics [70–74], extended versions of the model to anyonic statistics [75,76], spatial exponential decay of the interaction [77] and a supersymmetric version [78], along with mappings onto other models, such as attractive fermions [79], a BCS model [80–82], the Kardar-Parisi-Zhang (KPZ) model [83], directed polymers [84], three-dimensional black holes [85] or the Yang-Mills problem on a 2-sphere [86].

In this work, we focus on theoretical and mathematical issues associated with the analytical description of the ground state of the LL model. After so many years since the discovery of the closed-form system of equations by Lieb and Liniger, the explicit analytical solution of the model is still lacking. In particular, a wide range of experimentally relevant, intermediate repulsive interactions is hardly accessed analytically by perturbation theory. A complete analytical understanding of the ground state, whose importance is comparable to the solution of the 2D Ising model, would be a benchmark towards solutions at finite temperature or generalizations to multicomponent systems, where explicit results from integrability are scarce to date. The goal of this article is to provide analytical estimates for the ground state energy, excitation spectrum and other related observables covering the experimentally relevant regime of intermediate interaction strengths $\gamma \in [1,10]$ as defined in Eq. (3) below, whose accuracy would be comparable to the ones accessed by numerical methods. For this purpose, we start from the strongly interacting regime. We use a systematic method to obtain corrections to the TG limit up to high orders. Since this method is intrinsically limited to high interaction strength and the convergence with the order of expansion quite slow, we transpose the problem to number theory and guess a general, partially resummed structure. We study its accuracy by computing several observables and comparing the results with numerics, and approximations available in the literature. We conclude that our conjecture is valid in a considerably wider range of interactions than high-order asymptotic expansions. Then, we study the excitation spectrum, harder to access analytically. By a careful analysis of its symmetry properties, we find new analytical approximate expressions that involve quantites computed at equilibrium and compare them with numerics. We then find that the most accurate approximation is actually valid in a wide range of intermediate to strong interactions. Moreover, the accuracy of our methods enable us to study quantitatively the regime of validity of Tomonaga-Luttinger liquid theory in terms of the interaction strength. We show that the range of validity in momentum and energy increases with the coupling constant.

The structure of the paper is as follows: in Section 2 we introduce the main features of the model and briefly discuss the weakly interacting regime. Then, focusing on the energy, we systematically evaluate corrections to the regime of infinite interactions up to order 20 and make conjectures about a possible resummation, using comparison with the numerics as an

accuracy test. We compute various ground-state properties linked to this quantity and give results whose accuracy is unprecedented and already sufficient for all practical purposes. In Section 3, we focus on the excitation spectra of the LL model and derive a simple expression which is very accurate in the whole range of repulsive interactions. In Section 4, we give our conclusions and outlook. Several Appendices dwell deeper on the mathematical details; an exact mapping onto the classical physics problem of the circular plate capacitor is also discussed.

## 2 Ground-state energy and local correlations from analyticity: methods and illustrations

### 2.1 Model and Bethe Ansatz equations in the thermodynamic limit

The 1D quantum gas composed of $N$ identical spinless point-like bosons of mass $m$ with contact interactions, confined to a line of length $L$, is described by the Lieb-Liniger Hamiltonian $H^{LL}$ which reads [2]

$$H^{LL} = \sum_{i=1}^{N} \left[ -\frac{\hbar^2}{2m} \frac{\partial^2}{\partial x_i^2} + \frac{g_{1D}}{2} \sum_{j \neq i} \delta(x_i - x_j) \right], \tag{1}$$

where $\{x_i\}_{i \in \{1,\dots,N\}}$ label the positions of the bosons, $\hbar$ is the Planck constant divided by $2\pi$, $g_{1D}$ is an effective one-dimensional coupling constant which can be deduced from experimental parameters [42, 43] and $\delta$ is the Dirac function. We assume $g_{1D} > 0$, corresponding to repulsive interactions. In second-quantized form, the Hamiltonian reads [2]

$$H^{LL}[\hat{\psi}] = \frac{\hbar^2}{2m} \int_0^L dx \frac{\partial \hat{\psi}^\dagger}{\partial x} \frac{\partial \hat{\psi}}{\partial x} + \frac{g_{1D}}{2} \int_0^L dx \hat{\psi}^\dagger \hat{\psi}^\dagger \hat{\psi} \hat{\psi}, \tag{2}$$

where $\hat{\psi}$ is a bosonic field operator satisfying the canonical commutation relations with its Hermitian conjugate: $[\hat{\psi}(x), \hat{\psi}^\dagger(x')] = \delta(x-x')$, $[\hat{\psi}(x), \hat{\psi}(x')] = [\hat{\psi}^\dagger(x), \hat{\psi}^\dagger(x')] = 0$. The dimensionless coupling constant for this model, known as Lieb's parameter, reads

$$\gamma = \frac{m g_{1D}}{n_0 \hbar^2}, \tag{3}$$

where $n_0 = N/L$ is the mean linear density. For a given atomic species, the coupling constant is experimentally tunable by confinement-induced resonances, Feshbach resonances or control over the density $n_0$. The limit $\gamma \to +\infty$ yields the Tonks-Girardeau gas. In this regime, an exact mapping on a noninteracting spinless Fermi gas allows for full solution of the model, with arbitrary external potential [3].

For a uniform 1D Bose gas with periodic boundary conditions, corresponding to a ring geometry and ensuring translational invariance, the Bethe hypothesis allows to derive exact expressions for the ground state energy, excitations and static correlations of the system at arbitrary interaction strength. The procedure, called coordinate Bethe Ansatz, is well known for this system and widely explained in the literature, we refer to [2, 87–89] for details. At finite $N$, this yields a system of $N$ transcendental equations. In the thermodynamic limit, the latter reduces to a set of three equations, namely

$$g(z; \alpha) - \frac{1}{2\pi} \int_{-1}^{1} dy \frac{2\alpha g(y; \alpha)}{\alpha^2 + (y-z)^2} = \frac{1}{2\pi}, \tag{4}$$

where $g(z;\alpha)$ denotes the quasi-momentum distribution in reduced units, $z$ is the pseudo-momentum in reduced unit such that its maximal value is 1 and $\alpha$ is a non-negative parameter, in a one-to-one correspondence with the Lieb parameter $\gamma$ introduced above via a second equation,

$$\gamma \int_{-1}^{1} dy\, g(y;\alpha) = \alpha. \tag{5}$$

The third equation yields the dimensionless average ground-state energy per particle $e$, linked to the total energy $E_0$ by $e(\gamma) \equiv \frac{2m}{\hbar^2}\frac{E_0(\gamma)}{Nn_0^2}$, according to

$$e(\gamma) = \frac{\int_{-1}^{1} dy\, g(y;\alpha(\gamma))y^2}{[\int_{-1}^{1} dy\, g(y;\alpha(\gamma))]^3}. \tag{6}$$

Interestingly, Eq. (4) is decoupled from Eqs. (5)-(6), which is a specificity of the ground state [12]. Equation (4) is a homogeneous type II Fredholm integral equation with Lorentzian kernel [90]. In the following we will refer to it as the Lieb equation for simplicity. Prior to Lieb and Liniger's work, it had been observed that its solution yields the exact capacitance of a capacitor formed of two circular coaxial parallel plates. For more details on this interesting link with a standard problem of electrostatics, and approximation methods that are not considered in the main text, we refer the reader to Appendix A. We also refer to [91] for an historical review of attempts to exactly solve this problem, that has resisted more than a century of efforts from mathematicians and physicists alike.

In order to determine the ground-state energy of the Lieb-Liniger model, the most crucial step is to solve the Lieb equation (4). This was done numerically by Lieb and Liniger for a few values of $\alpha$ spanning several decades of interaction strengths [2]. The solution procedure relies on the following steps. An arbitrary positive value is fixed for $\alpha$, and Eq. (4) is solved, i.e. $g$ is found with the required accuracy as a function of $z \in [-1,1]$. Then, Eq. (5) yields $\gamma(\alpha)$, subsequently inverted to get $\alpha(\gamma)$. In doing so, one notices that $\gamma(\alpha)$ is an increasing function, thus interaction regimes are defined the same way for both variables. The energy is then easily obtained from Eq. (6), as well as many interesting observables, that are combinations of its derivatives. They all depend on the sole Lieb parameter, which is the key of the conceptual simplicity of the model.

In the following, we briefly tackle the weakly-interacting regime $\alpha \ll 1$, mostly for the sake of completeness, and dwell much deeper on the strongly-interacting regime $\alpha \gg 1$.

## 2.2 Expansions in the weakly-interacting regime

Both numerically and analytically, finding accurate approximate solutions of Eq. (4) at small values of the parameter $\alpha$ is a more involved task than at higher couplings, due to the singularity of the function $g$ at $\alpha = 0$, whose physical interpretation is that noninteracting bosons are not stable in 1D. A guess function was proposed by Lieb and Liniger [2], namely

$$g(z;\alpha) \simeq_{\alpha \ll 1} \frac{\sqrt{1-z^2}}{2\pi\alpha}, \tag{7}$$

which is a semi-circle law, rigorously derived in [92]. Heuristic arguments have suggested the following correction far from the edges in the variable $z$ [92, 93]:

$$g(z;\alpha) \simeq_{\alpha \ll 1, \alpha \ll |1 \pm z|} \frac{\sqrt{1-z^2}}{2\pi\alpha} + \frac{1}{4\pi^2\sqrt{1-z^2}}\left[ z\ln\left(\frac{1-z}{1+z}\right) + \ln\left(\frac{16\pi}{\alpha}\right) + 1 \right], \tag{8}$$

reproduced later by direct calculation in [94] after regularization of divergent series. A technical difficulty consists in evaluating precisely the validity range of both approximations Eq.

(7) and Eq. (8) in $\alpha$, and whether the latter offers a significant improvement in accuracy. Systematic comparison to numerical solutions at $\alpha \ll 1$ is beyond the scope of this work. One can nevertheless, to a certain extent, discuss the relative accuracy of two solutions at a given value of $\alpha$, using the methods detailed in Appendix B.

The highest-order exact expansion proven to date for the dimensionless energy in the weakly interacting regime is

$$e_{exact}(\gamma) = \gamma - \frac{4}{3\pi}\gamma^{3/2} + \left[\frac{1}{6} - \frac{1}{\pi^2}\right]\gamma^2 - D\gamma^{5/2} + O(\gamma^3). \tag{9}$$

The first order coefficient is readily obtained from Eq. (7), the second order one is found using Eq. (8) and coincides with the result from Bogoliubov's approximation [2]. The prefactor of the $\gamma^2$ term has been more controversial and required important efforts. Its expression was inferred on numerical grounds in [95], while in [96] a factor $1/8$ instead of $1/6$ is found analytically after lengthy calculations. It seems, though, that the factor $1/6$ is correct, as it was recently recovered from potential theory in [97] in a (quasi-)rigorous way, and coincides with numerical independent calculations carried out in [98]. The exact fourth term, given by multiple integrals, was numerically evaluated to $D \simeq 0.0016$ [99]. A similar value was then found by fitting using very accurate numerics [100].

Our contribution to the problem is purely numerical in this regime. We evaluated the function $e(\gamma)$ numerically by solving the Bethe Ansatz equations with a Monte-Carlo algorithm for 17 values of the interaction strength spanning the interval $\gamma \in [1, 15]$. A fit containing the known analytical prefactors and four adjustable coefficients yields

$$e_{fit}(\gamma) = \gamma - \frac{4}{3\pi}\gamma^{3/2} + \left[\frac{1}{6} - \frac{1}{\pi^2}\right]\gamma^2 - 0.002005\gamma^{5/2} +$$
$$0.000419\gamma^3 - 0.000284\gamma^{7/2} + 0.000031\gamma^4 \tag{10}$$

with a relative error as low as a few per thousands in the interval considered. We stress that in this expansion the various coefficients are not supposed to (and do not) coincide with the exact ones in the Taylor expansion. The expression yields, however, a rather good estimate for the ground-state energy from the noninteracting to the strong-coupling regime for many practical purposes. In particular, it will allow us to check whether the results of next section, where this regime is specifically addressed, are also valid at intermediate or even weak interactions.

## 2.3 Expansions in the strongly-interacting regime

Close to the Tonks-Girardeau regime, accurate asymptotic solutions of the Lieb equation Eq. (4) can be found using a double series expansion of the $g$ function in $z$ and $1/\alpha$ around $(0,0)$. Our starting point is a slightly simplified version of a recently-developed procedure [1], we refer to Appendix C for a detailed derivation and mathematical discussions. This systematic expansion scheme is valid for $\alpha > 2$ and yields an approximate solution of the form

$$g(z; \alpha, M) = \sum_{k=0}^{2M+2} \sum_{j=0}^{M} g_{jk} \frac{z^{2j}}{\alpha^k}, \tag{11}$$

where $M$ is an integer cutoff, and $g_{jk}$, by construction, are polynomials of $1/\pi$ with rational coefficients. This expansion coincides with the Taylor expansion at the chosen order, and $g(z; \alpha, M)$ converges to $g(z; \alpha)$ for $M \to +\infty$.

We stress that the lowest interaction strength attainable with the strong coupling expansion, i.e. $\alpha = 2$, corresponds to an interaction strength $\gamma = \gamma_c \simeq 4.527$ [1]. This is low enough to combine with the result in the weakly-interacting regime, and obtain an accurate

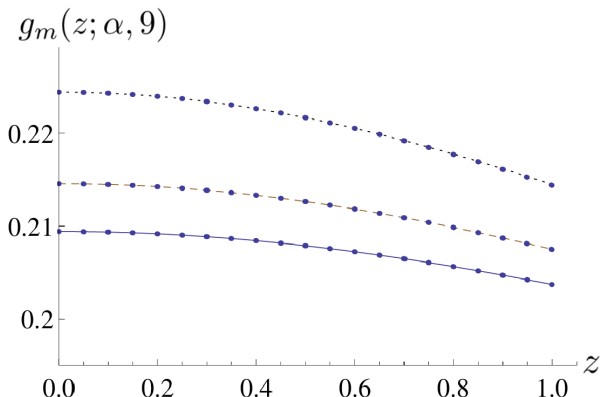

Figure 1: Dimensionless function $g_m(z; \alpha, 9)$, mean of the $18^{th}$ and $20^{th}$ order in $1/\alpha$ of $g(z; \alpha)$, as a function of the dimensionless variable $z$, at dimensionless parameters $\alpha = 2.5$ (solid, blue), $\alpha = 2.3$ (dashed, brown) and $\alpha = 2$ (dotted, black) from bottom to top, compared to the corresponding numerically exact solutions (blue dots). Only a few numerical values are shown to improve the visibility, and numerical error is within the size of the dots.

description of the ground state of the model over the whole range of repulsive interactions. Furthermore, this method yields several orders of perturbation theory at each step and is automatically consistent at all orders. Nonetheless, it is crucial to capture the correct behavior of $g$ as a function of $z$ in the whole interval $[-1, 1]$ to obtain accurate expressions, whereas the expansion converges slowly to the exact value at the extremities since the Taylor expansion in $z$ is done around the origin. This reflects in the fact that the maximum exponent of $z^2$ varies more slowly with $M$ than the one of $1/\alpha$. All in all, this drawback leads to a strong limitation of the validity range in $1/\alpha$ at a given order, and calls for high order expansions, far beyond the maximal order treated so far, $M = 3$ [1]. For a detailed study of the accuracy of the method in terms of the cutoff $M$, we refer to Appendix C. The high number of corrections needed seems at first redhibitory, but we noticed that doing the average over two consecutive orders in $M$, denoted by

$$g_m(z; \alpha, M) = \frac{g(z; \alpha, M) + g(z; \alpha, M-1)}{2}, \tag{12}$$

dramatically increases the accuracy, yielding an excellent agreement with numerical calculations for all $\alpha \in [2, +\infty[$ at $M = 9$, as illustrated in Fig. 1.

However, at increasing $M$ the method quickly yields too unhandy expressions for the function $g$, as it generates $1 + (M+1)(M+2)(M+3)/3$ terms. This motivated us to seek compact representations and resummations for the function $g(z; \alpha, M)$, allowing to easily use them for further applications and to generate them up to large orders. We present below an analysis of the structure of the terms entering the expansion (11) and a conjecture for compact expressions, yielding a partial resummation. Our conjecture has been then verified and validated by a very recent numerical approach [101, 102].

## 2.4 Conjectural expansions and resummations in the strongly-interacting regime

### 2.4.1 Conjectures for $g(z; \alpha)$

By a careful analysis of the terms of the series expansion, we found two apparently distinct groups of patterns, which we arbitrarily call 'first kind' and 'second kind' respectively. Terms

of the first kind already arise at low orders in $z$ and $1/\alpha$, while terms of the second kind appear at higher orders and are expected to play a crucial role in the crossover region of intermediate interactions. These structures are conjectural, we infered them on as few first terms as possible and systematically checked that their predictions coincide with all higher-order available terms. While the simplest patterns are trivial to figure out, others are far more difficult to find because of their increasing structural complexity. Denoting by $m$ the 'kind' and by $n$ an index for the elusive notion of 'complexity', we write

$$g(z; \alpha, M) = \frac{1}{2\pi} + \sum_{m,n} I_{m,n}^M(z; \alpha) , \qquad (13)$$

where each $I_{m,n}^M$ is itself a double sum over all terms of given kind $m$ and complexity $n$ that appear at order $M$. As an illustration, we detail the terms found to all orders up to $M = 9$ included in Eq. (11).

The simplest term is

$$I_{1,0}^M = \frac{1}{\pi^2 \alpha} \sum_{j=0}^{M} (-1)^j \left(\frac{z}{\alpha}\right)^{2j} \sum_{k=0}^{2(M-j)+1} \left(\frac{2}{\pi\alpha}\right)^k . \qquad (14)$$

Terms of the first kind with complexity $n = 1$ sum as

$$I_{1,1}^M = -\frac{1}{3\pi^2 \alpha^3} \sum_{j=0}^{M-1} (-1)^j \left(\frac{z}{\alpha}\right)^{2j} \sum_{k=0}^{2(M-j)-1} \left(\frac{2}{\pi\alpha}\right)^k [2k + (j+1)(2j+1)]. \qquad (15)$$

Terms of the first kind and complexity $n = 2$ are

$$I_{1,2}^M = \frac{1}{45\pi^3 \alpha^6} \sum_{j=0}^{M-2} (-1)^j \left(\frac{z}{\alpha}\right)^{2j} \sum_{k=0}^{2(M-j)-4} \left(\frac{2}{\pi\alpha}\right)^k (20k^2 + a_j k + b_j), \qquad (16)$$

where $a_j = 4 * (10j^2 + 15j + 36)$ and $b_j = 12j^4 + 60j^3 + 161j^2 + 159j + 142$. Terms of the first kind and complexity $n = 3$ are

$$I_{1,3}^M = -\frac{1}{2835\pi^3 \alpha^8} \sum_{j=0}^{M-3} (-1)^j \left(\frac{z}{\alpha}\right)^{2j} \sum_{k=0}^{2(M-j)-6} \left(\frac{2}{\pi\alpha}\right)^k (280k^3 + c_j k^2 + d_j k + e_j) \qquad (17)$$

where $c_j = 84 * (10j^2 + 15j + 57)$, $d_j = 2 * (252j^4 + 1260j^3 + 5145j^2 + 5985j + 11476)$ and $e_j = 72j^6 + 756j^5 + 3942j^4 + 10575j^3 + 21150j^2 + 19287j + 18414$. The only term of the first kind and complexity $n = 4$ we have identified is

$$\frac{2}{42525} \frac{1}{\pi^3 \alpha^{10}} \sum_{k=0}^{2M-8} \left(\frac{2}{\pi\alpha}\right)^k (350k^4 + 10920k^3 + 118372k^2 + 474672k + 334611), \qquad (18)$$

which should correspond to the terms with index $j = 0$ in $I_{1,4}^M$. Note that, for terms of first kind, complexity actually corresponds to the degree of the polynomial in $k$.

Terms of the second kind and lowest complexity are

$$I_{2,0}^M = \frac{1}{\pi^2 \alpha^5} \sum_{j=0}^{M-2} \frac{(-1)^j}{(2j+5)\alpha^{2j}}. \qquad (19)$$

We also found

$$I_{2,1}^M = -\frac{1}{2} \frac{1}{\pi^2 \alpha^5} \left(\frac{z}{\alpha}\right)^2 \sum_{j=0}^{M-3} \left(\frac{z}{\alpha}\right)^{2j} \frac{(-1)^j}{j+1} \sum_{k=0}^{M-j-3} \frac{(-1)^k}{\alpha^{2k}} \binom{2(k+j+3)}{2j+1}. \qquad (20)$$

We note that some terms of the first kind could be interpreted as second kind, thus involving a shift of the summation index in one of them, so that the proposed classification may not be optimal. However, if one performs the summation of all terms explicited here, the resulting expansion is exact up to order $1/\alpha^{11}$ included, thus the proposed structures are efficient to encode many terms of the series in a relatively compact way. By comparing with high-precision numerical calculations, we find that for $\alpha = 2$ the expansion has an error of the order of $1-2\%$. The problem of the full resummation of the series expansion for $g(z;\alpha)$ remains open.

### 2.4.2 Conjectures for $e(\gamma)$

In this section, we focus on resummation patterns directly on the dimensionless ground-state energy $e(\gamma)$. Here, we shall write $e(\gamma) = \sum_{n=0}^{+\infty} e_n(\gamma)$, where once again the index $n$ denotes a notion of complexity. Focusing on the large-$\gamma$ asymptotic expansion, we identify the pattern of a first sequence of terms. We conjecture that they appear at all orders and resum the series, obtaining

$$\frac{e_0(\gamma)}{e^{TG}} = \sum_{k=0}^{+\infty} \frac{(-1)^k 2^k \binom{k+1}{1}}{\gamma^k} = \frac{\gamma^2}{(2+\gamma)^2}, \tag{21}$$

where $e^{TG} = \pi^2/3$ is the value in the Tonks-Girardeau regime. Then, using the expansion of $e(\gamma)$ in $1/\gamma$ up to $M = 9$ (given in Appendix E with numerical coefficients) and guided by the property

$$\sum_{k=0}^{+\infty} \frac{(-1)^k 2^k \binom{k+3n+1}{3n+1}}{\gamma^k} = \left(\frac{\gamma}{\gamma+2}\right)^{3n+2}, \tag{22}$$

we conjecture that the structure of the term of complexity $n \geq 1$ defined above is

$$\frac{e_n(\gamma)}{e^{TG}} = \frac{\pi^{2n}\gamma^2 \mathscr{L}_n(\gamma)}{(2+\gamma)^{3n+2}}, \tag{23}$$

where $\mathscr{L}_n$ is a polynomial of degree $n-1$, whose coefficients are rational, non-zero and of alternate signs. In this context, the notion of complexity is directly related to the power of the denominator. The first few polynomials are found as

$$\mathscr{L}_1(\gamma) = \frac{32}{15},$$
$$\mathscr{L}_2(\gamma) = -\frac{96}{35}\gamma + \frac{848}{315},$$
$$\mathscr{L}_3(\gamma) = \frac{512}{105}\gamma^2 - \frac{4352}{525}\gamma + \frac{13184}{4725},$$
$$\mathscr{L}_4(\gamma) = -\frac{1024}{99}\gamma^3 + \frac{131584}{5775}\gamma^2 - \frac{4096}{275}\gamma + \frac{11776}{3465},$$
$$\mathscr{L}_5(\gamma) = \frac{24576}{1001}\gamma^4 - \frac{296050688}{4729725}\gamma^3 + \frac{453367808}{7882875}\gamma^2 - \frac{227944448}{7882875}\gamma + \frac{533377024}{212837625},$$
$$\mathscr{L}_6(\gamma) = -\frac{4096}{65}\gamma^5 + \frac{6140928}{35035}\gamma^4 - \frac{4695891968}{23648625}\gamma^3 + \frac{3710763008}{23648625}\gamma^2 - \frac{152281088}{4729725}\gamma + \frac{134336512}{42567525}$$
$$\tag{24}$$

by identification with the $1/\gamma$ expansion to order 20. We conjecture that the coefficient of the highest-degree monomial of $\mathscr{L}_n$ is $\frac{3*(-1)^{n+1}*2^{2n+3}}{(n+2)(2n+1)(2n+3)}$.

Interestingly, contrary to the $1/\gamma$ expansion, those partially resummed terms are not divergent at small $\gamma$, increasing the validity range. We also notice that $e_0$ corresponds to Lieb

and Liniger's approximate solution assuming a uniform density of pseudo-momenta [2], and an equation equivalent to Eq. (21) appears in [103]. The first correction $e_1$ was predicted rigorously in [96], thus supporting our conjectures.

In the next section, we will evaluate the quality of our conjecture (23), (24) by comparing with high-precision numerical calculations.

## 2.5 High-precision ground-state properties

In this section we present our predictions for various physical quantities, using a combination of weak-coupling and strong-coupling expansion as well as the conjectures.

### 2.5.1 Density of pseudo-momenta

The density of pseudo-momenta follows immediately from the solution of the Lieb equation (4). It is defined as

$$\rho(k; \gamma) = g(Q(\gamma)z; \alpha(\gamma)), \tag{25}$$

where $k$ denote the pseudo-momenta, whose maximal value is $Q(\gamma)$, such that [87]

$$\frac{Q(\gamma)}{k_F} = \frac{1}{\pi \int_{-1}^{1} dz\, g(z; \alpha(\gamma))}, \tag{26}$$

where $k_F = \pi n_0$ is the Fermi wavevector in 1D. Within the method explained in Appendix C, we have access to analytical expressions for $\alpha > 2$ only. For a more general method to obtain this function, valid for any $\alpha$, we refer to Appendix D. Since the latter method is not appropriate to obtain analytical expressions of other quantities, we do not dwell further on it in the main text. Figure 2 shows our results for the density of pseudo-momenta. In the Tonks-Girardeau regime $\gamma = +\infty$ the density of pseudo-momenta coincides with the Fermi distribution at zero temperature, which is a manifestation of the effective fermionization. At decreasing interactions away from the Tonks-Girardeau gas, we find that the distribution remains quasi-uniform in a wide range of strong interactions. This shows the robustness of the effective Fermi-like structure, yet with Fermi wavevector which is progressively renormalized. Then, at lower interaction strengths, we witness an increase of the height of the peak of the distribution, which becomes progressively sharper and narrower around the origin in the weakly-interacting, quasi-condensate regime.

### 2.5.2 Ground-state energy

We show in Fig. 3 the dimensionless ground-state energy per particle over a wide range of repulsive interactions. Our expressions in both the weakly-interacting regime as given by Eq. (10) and the strongly-interacting regime as given by Eq. (23) are compared to the numerics to emphasize their accuracy. As main result, we find that they have a wide overlap in the regime of intermediate interactions and are hard to distinguish from the numerics. In particular, extrapolating the conjecture in the strongly-interacting regime to low values of $\gamma$, we obtain a substantial improvement compared to the previously known approximations $e_0$ and $e_0 + e_1$ in Eqs. (21) and (23) when $\gamma \gtrsim 1$.

We then show our results for the ratios of the mean kinetic energy $e_k$ and interaction energy $e_p$ per particle. According to Pauli's theorem [2], they are obtained as

$$e_p(\gamma) = \gamma \frac{de}{d\gamma} \tag{27}$$

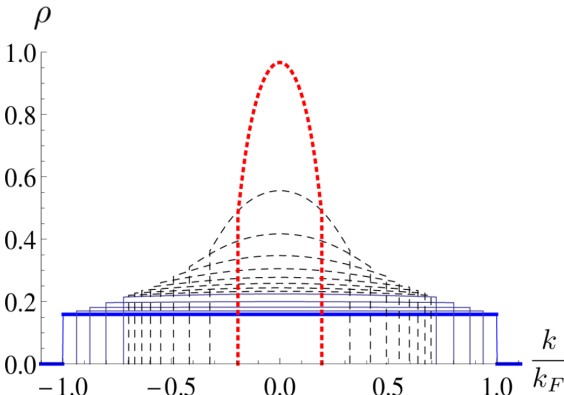

Figure 2: Dimensionless density of pseudo-momenta $\rho$ as a function of dimensionless pseudo-momentum $k/k_F$ for various interaction strengths. Different colors and line styles represent results from various approximations. From bottom to top, one sees the exact result in the Tonks-Girardeau regime (blue, thick), then four curves corresponding to dimensionless parameters $\alpha = 10, 5, 3$ and $2$ respectively (solid, blue) obtained from the analytical methods of Appendices C, D and a Monte-Carlo algorithm to solve the Lieb equation Eq. (4) (indistinguishable from each other). Above, an other set of curves represents interaction strengths from $\alpha = 1.8$ to $\alpha = 0.4$ with step $-0.2$ (black, dashed) obtained from a Monte-Carlo algorithm and the method of Appendix D, where again analytics and numerics are indistinguishable. Finally, we also plotted the results at $\alpha = 0.2$ from the method of Appendix D (dotted, red).

and

$$e_k(\gamma) = \left(e - \gamma \frac{de}{d\gamma}\right). \tag{28}$$

These quantities are shown in the left panel of Fig. 4, normalized to the total energy in the Tonks-Girardeau regime as in [104]. The kinetic energy is maximal in the Tonks-Girardeau regime of ultra-strong interactions. This can be seen as a manifestation of fermionization, since in several respects the particles behave as free fermions in this limit due to the Bose-Fermi mapping [3]. The right panel of Fig. 4 shows the ratio of interaction to kinetic energy. This quantity scales as $\gamma^{-1/2}$ in the weakly-interacting regime and decreases monotonically at increasing $\gamma$, thus showing that $\gamma$ does not represent this ratio, contrary to the mean-field prediction.

### 2.5.3 Local correlation functions

In experiments, it is possible to access to the local $k$-particle correlation functions $g_k$ of the Lieb-Liniger model, defined as

$$g_k = \frac{\langle [\hat{\psi}^\dagger(0)]^k [\hat{\psi}(0)]^k \rangle}{n_0^k}, \tag{29}$$

where $\langle . \rangle$ represents the ground-state average.

The local pair correlation $g_2$ (respectively three-body correlation $g_3$) is a measure of the probability of observing two (three) particles at the same position. In particular, $g_3$ governs the rates of inelastic processes, such as three-body recombination and photoassociation in pair collisions. The second-order correlation, $g_2$, is easily obtained with our method using the Hellmann-Feynman theorem, that yields $g_2 = \frac{de}{d\gamma}$ [105]. Hence, the fact that $e(\gamma)$ is an

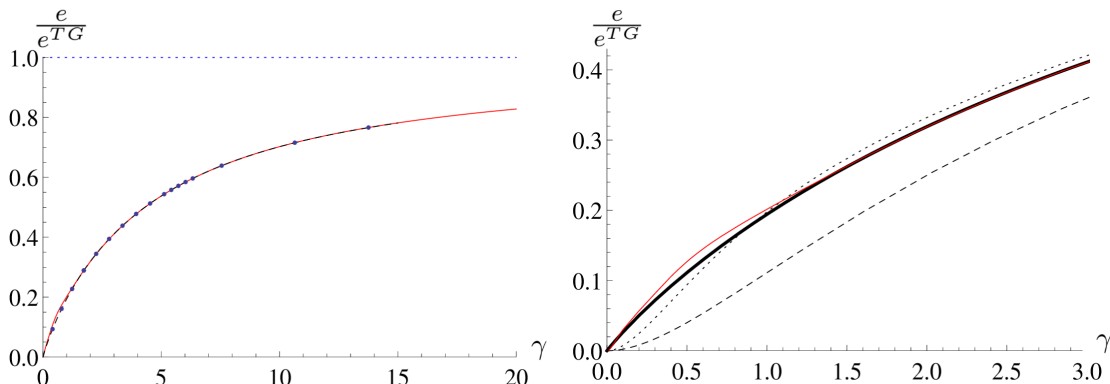

Figure 3: Left panel: dimensionless ground state energy per particle $e$ normalized to its value in the Tonks-Girardeau limit $e^{TG}$ (dotted, blue), as a function of the dimensionless interaction strength $\gamma$: conjectural expansion at large $\gamma$ (solid, red) as given by Eq. (23) to sixth order, small $\gamma$ expansion (black, dashed) as given by Eq. (10) and numerics (blue points). Right panel: zoom in the weakly-interacting region. Numerically exact result (black, thick) is compared to $e_0$ (black, dashed), $e_0 + e_1$ (black, dotted) and the sixth-order expansion (red) in Eq. (23).

increasing function is actually a direct consequence of the positiveness of $g_2$. Higher-order correlation functions are related in a non-trivial way to the moments of the density of pseudo-momenta, defined as

$$\epsilon_k(\gamma) \equiv \frac{\int_{-1}^{1} dz z^k g(z; \alpha(\gamma))}{[\int_{-1}^{1} dz g(z; \alpha(\gamma))]^{k+1}}. \tag{30}$$

In particular, $g_3$ is related to the two first non-zero moments by the relation [106]

$$g_3(\gamma) = \frac{3}{2\gamma} \frac{d\epsilon_4}{d\gamma} - \frac{5\epsilon_4}{\gamma^2} + \left(1 + \frac{\gamma}{2}\right) \frac{d\epsilon_2}{d\gamma} - 2\frac{\epsilon_2}{\gamma} - 3\frac{\epsilon_2}{\gamma} \frac{d\epsilon_2}{d\gamma} + 9\frac{\epsilon_2^2}{\gamma^2}. \tag{31}$$

By definition, the second moment $\epsilon_2$ coincides with the dimensionless energy $e$. In Fig. 5 we plot accurate expressions for $g_3$ obtained in [107], and our analytical expression of $g_2$ readily obtained from Eqs. (10) and (23). The fact that $g_2$ vanishes in the Tonks-Girardeau regime is once again a consequence of fermionization, as interactions induce a kind of Pauli principle and preclude that two bosons come in contact. This property has been the key to realize the TG gas experimentally [105]. At very small interaction strengths, however, the ratio $g_3/g_2$ can not be neglected; this means that the weakly-interacting 1D Bose gas is less stable with respect to three-body losses than the strongly-interacting one.

### 2.5.4 Non-local, one-body correlation function and Tan's contact

Finally, we study the one-body non-local correlation function $g_1(x) \equiv \langle \hat{\psi}^\dagger(x)\hat{\psi}(0)\rangle / n_0$. We focus first on the Tonks-Girardeau regime [108–110], where expansions at short and long distances are now known to high enough orders to match at intermediate distances, as can be seen in Fig. 6. We use the notation $z = k_F x$, where $k_F = \pi n_0$ is the Fermi wavevector in 1D. We recall first the large-distance expansion derived in [110](with signs of the coefficients

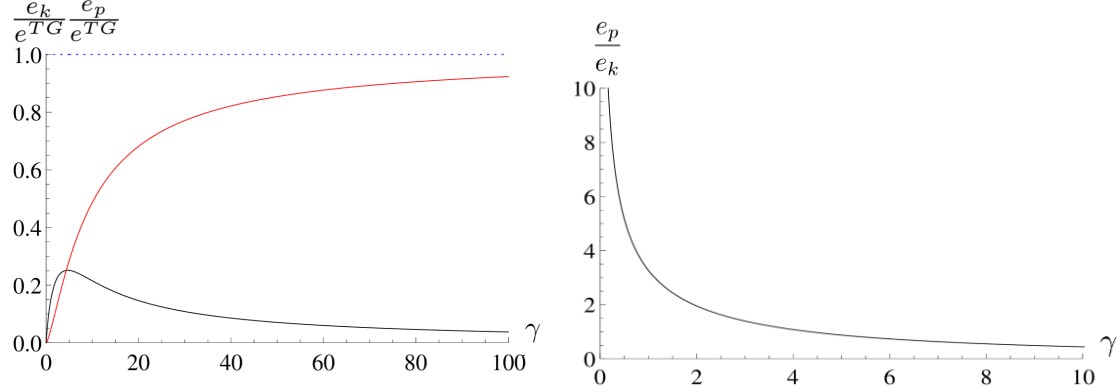

Figure 4: Left panel: dimensionless ground-state kinetic energy per particle (red) and interaction energy per particle (black), normalized to the total energy per particle in the Tonks-Girardeau limit $e^{TG}$, as a function of the dimensionless interaction strength $\gamma$. The horizontal line (blue, dotted) is a guide to the eye. Right panel: dimensionless ratio of the interaction and kinetic energy as a function of $\gamma$.

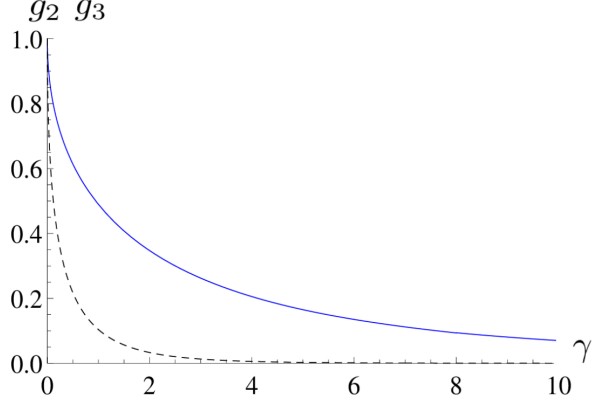

Figure 5: Dimensionless correlation functions $g_2$ (blue, solid) from Eqs. (10) and (24) and $g_3$ from Ref. [107] (black, dashed) as functions of the dimensionless interaction strength $\gamma$. Three-body processes are strongly suppressed at high interaction strength but become of the same order of magnitude as two-body processes in the quasi-condensate regime.

corrected in [111])

$$g_1^{TG}(z) = \frac{G(3/2)^4}{\sqrt{2|z|}} \left[ 1 - \frac{1}{32z^2} - \frac{\cos(2z)}{8z^2} - \frac{3}{16}\frac{\sin(2z)}{z^3} + \frac{33}{2048}\frac{1}{z^4} + \frac{93}{256}\frac{\cos(2z)}{z^4} + O\left(\frac{1}{z^5}\right) \right],$$
(32)

where $G$ is the Barnes function defined as $G(1) = 1$ and the functional relation $G(z+1) = \Gamma(z)G(z)$, $\Gamma$ being the Euler Gamma function.

At short distances, using the same technique as in [112] to solve the sixth Painlevé equa-

tion, we find the following expansion, where we added six orders compared to [110]:

$$g_1^{TG}(z) = \sum_{k=0}^{8} \frac{(-1)^k z^{2k}}{(2k+1)!} + \frac{|z|^3}{9\pi} - \frac{11|z|^5}{1350\pi} + \frac{61|z|^7}{264600\pi} + \frac{z^8}{24300\pi^2} - \frac{253|z|^9}{71442000\pi} - \frac{163z^{10}}{59535000\pi^2}$$
$$+ \frac{7141|z|^{11}}{207467568000\pi} + \frac{589z^{12}}{6429780000\pi^2} - \frac{113623|z|^{13}}{490868265888000\pi} - \frac{2447503z^{14}}{1143664968600000\pi^2}$$
$$+ \left( \frac{1}{40186125000\pi^3} + \frac{33661}{29452095953280000\pi} \right) |z|^{15} + \frac{5597693}{140566821595200000\pi^2} z^{16} + O(|z|^{17}).$$
$$(33)$$

The first sum is a truncation of the integer series defining the function $\sin(z)/z$, which corresponds to the one-body correlation function of noninteracting fermions. The additional terms appearing in this expansion, and in particular the odd ones, are peculiar of bosons with contact interactions. Actually, the one-body correlation function for Tonks-Girardeau bosons differs from the one of a Fermi gas due to the fact that it is a nonlocal observable, depending also on the phase of the wavefunction and not only on its modulus.

The same structure is valid at finite interaction strength, where the short-distance expansion reads

$$g_1(z) = 1 + \sum_{i=1}^{+\infty} \frac{c_i}{\pi^i} |z|^i , \tag{34}$$

and the first coefficients are explicitly found as [113]

$$c_1 = 0,$$
$$c_2 = -\frac{1}{2} e_k,$$
$$c_3 = \frac{1}{12} \gamma^2 \frac{de}{d\gamma}, \tag{35}$$

and [114, 115]

$$c_4 = \frac{\gamma}{12} \frac{d\epsilon_4}{d\gamma} - \frac{3\epsilon_4}{8} + \frac{2\gamma^2 + \gamma^3}{24} \frac{de}{d\gamma} - \frac{\gamma e}{6} - \frac{\gamma}{4} e \frac{de}{d\gamma} + \frac{3}{4} e^2. \tag{36}$$

Our solution of the Lieb equation Eq. (4) hence allows to estimate the first terms of this expansion. Further progress on analytic expressions has been obtained in [113, 116], as well as numerically [35, 117].

The Fourier transform of the one-body correlation function is the momentum distribution of the gas,

$$n(k) = n_0 \int_{-\infty}^{+\infty} dx\, g_1(x) e^{-ikx}. \tag{37}$$

The short-distance behavior of $g_1$ in Eq. (34) allows to obtain the large momenta asymptotic behavior,

$$\frac{k^4 n(k)}{n_0^4} \xrightarrow[k \to +\infty]{} \mathscr{C}, \tag{38}$$

where $\mathscr{C}(\gamma) = \gamma^2 \frac{de}{d\gamma}$ [113, 118] is Tan's contact [119]. Figure 7 shows the value of Tan's contact obtained from Eqs. (10) and (23)-(24).

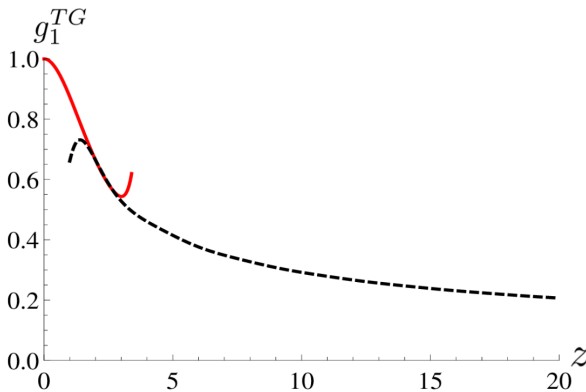

Figure 6: Dimensionless one-body correlation function $g_1^{TG}$ in the Tonks-Girardeau regime as a function of the dimensionless distance $z$. Short-distance asymptotics given by Eq. (33) (red, solid) and long-distance asymptotics given by Eq. (32) (black, dashed).

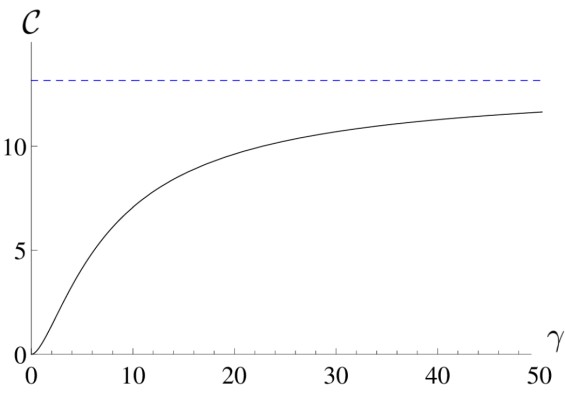

Figure 7: Dimensionless Tan's contact $\mathscr{C}$ as a function of dimensionless interaction strength $\gamma$ (solid, black) and its value in the Tonks-Girardeau limit, $\mathscr{C}^{TG} = 4\pi^2/3$ (dashed, blue).

## 3  Excitation spectrum, exact and approximate results

Now that the ground-state energy of the model is known with good accuracy, we proceed with a more complicated and partially open problem, namely the analytical characterization of excitations above the ground state at zero temperature. In particular, we are interested in the accuracy of field-theoretical approximations such as Luttinger liquid theory. In order to introduce these excitations, we consider the Bethe Ansatz solution at finite $N$. The total momentum $P$ and energy $E$ of the system are given by $P = \hbar \sum k_j$ and $E = \frac{\hbar^2}{2m} \sum k_j^2$ respectively [4], where the set of quasi-momenta $\{k_j\}$ satisfies the following system of $N$ transcendental equations

$$\left\{ k_j = \frac{2\pi}{L} I_j - \frac{2}{L} \sum_{l=1}^{N} \arctan\left( \frac{k_j - k_l}{\gamma n_0} \right) \right\}_{j \in \{-\frac{N-1}{2}, \ldots, \frac{N-1}{2}\}} . \tag{39}$$

The $I_j$'s, called Bethe quantum numbers, are integer for odd values of $N$ and half-odd if $N$ is even. Since we consider $\gamma > 0$, the quasi-momenta are real and can be ordered in such a way that $k_1 < k_2 < \cdots < k_N$. Then, automatically $I_1 < I_2 < \cdots < I_N$ [4]. The ground state corresponds to $I_j = -\frac{N+1}{2} + j$ and has total momentum $P_{GS} = 0$ at arbitrary interaction

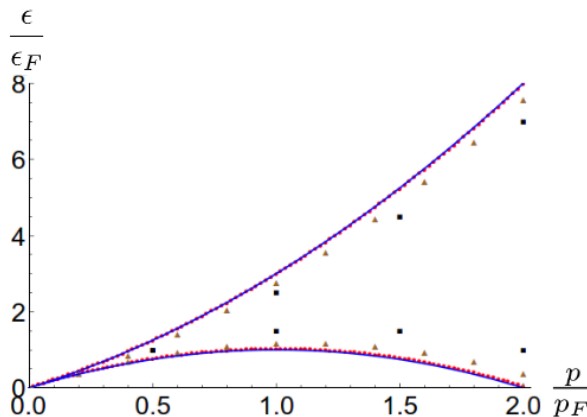

Figure 8: Excitation energy of the Tonks-Girardeau gas $\epsilon$ in units of the Fermi energy in the thermodynamic limit $\epsilon_F = \frac{\hbar^2\pi^2 N^2}{2mL^2}$, as a function of the excitation momentum $p$, in units of the the Fermi momentum in the thermodynamic limit $p_F = \frac{\hbar\pi N}{L}$, for $N = 4$ (black squares), $N = 10$ (brown triangles), $N = 100$ (red dots). The last one is quasi-indistinguishable from the excitation spectra in the thermodynamic limit (solid, blue).

strength. In what follows, we use the notations $p = P - P_{GS}$ and $\epsilon = E - E_{GS}$ for the total momentum and energy of an excitation with respect to the ground state, so that the excitation spectrum is defined as $\epsilon(p)$. For symmetry reasons, we will only consider excitations such that $p \geq 0$, those with $-p$ having the same energy.

The Lieb-Liniger model features two excitation branches, denoted as type I and type II [4]. In order to explain their features, we derive them in the Tonks-Girardeau limit [120], where the set of equations (39) decouples and $k_j = \frac{2\pi}{L} j$. The type-I excitations corresponding to the highest energy excitation, occur when the highest-energy particle with $j = (N-1)/2$ gains a momentum $p_n = \hbar 2\pi n/L$ and an energy $\epsilon_n^I = \frac{\hbar^2\pi^2}{2mL^2}[(N-1+2n)^2 - (N-1)^2]$. The corresponding dispersion relation is

$$\epsilon^I(p) = \frac{1}{2m}\left[ 2p_F p\left(1 - \frac{1}{N}\right) + p^2 \right] \tag{40}$$

where $p_F = \pi\hbar N/L$ is the Fermi momentum.

The type-II excitations occur when a particle inside the Fermi sphere is excited to occupy the lowest energy state available, carrying a momentum $p_n = 2\pi\hbar n/L$. This type of excitation corresponds to shifting all the rapidities with $j' > n$ by $2\pi\hbar/L$, thus leaving a hole in the Fermi sea. This corresponds to an excitation energy $\epsilon_n^{II} = \frac{\hbar^2\pi^2}{2mL^2}[(N+1)^2 - (N+1-2n)^2]$, yielding the excitation branch

$$\epsilon^{II}(p) = \frac{1}{2m}\left[ 2p_F p\left(1 + \frac{1}{N}\right) - p^2 \right]. \tag{41}$$

Any combination of one-particle and one-hole excitation is possible, giving rise to an intermediate energy between $\epsilon^I(p)$ and $\epsilon^{II}(p)$, forming a continuum in the thermodynamic limit. Figure 8 shows the type-I and type-II excitation spectrum for bosons in the Tonks-Girardeau limit. We notice the symmetry $p \leftrightarrow 2p_F - p$, valid at large boson number for the type-II branch.

In the general case of finite interaction strength, the system of equations (39) for the many-body problem can not be solved analytically, although expansions in the interaction strength can be obtained in the weakly- and strongly-interacting regimes [71]. The solution is easily obtained numerically for a few bosons. To reach the thermodynamic limit with several digits accuracy the Tonks-Girardeau treatment suggests that $N$ should be of the order of 100, and the

interplay between interactions and finite $N$ may slow down the convergence at finite $\gamma$ [34], but a numerical treatment is still possible. Here, we shall directly address the problem in the thermodynamic limit, where it reduces to two equations [1, 121] :

$$p(k;\gamma) = 2\pi\hbar Q(\gamma) \left| \int_1^{k/Q(\gamma)} dy\, g(y;\alpha(\gamma)) \right| \tag{42}$$

and

$$\epsilon(k;\gamma) = \frac{\hbar^2 Q^2(\gamma)}{m} \left| \int_1^{k/Q(\gamma)} dy\, f(y;\alpha(\gamma)) \right|, \tag{43}$$

where, according to Eq. (26), $Q(\gamma) = n_0/[\int_{-1}^1 dy\, g(y;\alpha(\gamma))]$. It is known as the Fermi rapidity, represents the radius of the quasi-Fermi sphere and equals $k_F$ in the Tonks-Girardeau regime. The function $f$ satisfies the integral equation

$$f(z;\alpha) - \frac{1}{\pi} \int_{-1}^1 dy \frac{\alpha}{\alpha^2 + (y-z)^2} f(y;\alpha) = z, \tag{44}$$

referred to as the second Lieb equation in what follows. We solve it with similar techniques as for the Lieb equation (4). Details are given in Appendix F.

The excitation spectra at a given interaction strength $\gamma$ are obtained in a parametric way as $\epsilon(k;\gamma)[p(k;\gamma)], k \in [0, +\infty[$. Within this representation, one can interpret the type I and type II spectra as a single curve, where the type I part corresponds to $|k|/Q \geq 1$ and thus to quasi-particle excitations, while type II is obtained for $|k|/Q \leq 1$, thus from processes taking place inside the quasi-Fermi sphere, which confirms that they correspond to quasi-hole excitations. Using basic algebra on Eqs. (42) and (43) we obtain the following interesting general results:

(i) The ground state ($p = 0, \epsilon = 0$) trivially corresponds to $k = Q(\gamma)$, showing that $Q$ represents the edge of the Fermi surface.

(ii) The quasimomentum $k = -Q(\gamma)$ corresponds to the umklapp point ($p = 2p_F, \epsilon = 0$), always reached by the type II spectrum in the thermodynamic limit, regardless of the value of $\gamma$.

(iii) The maximal excitation energy associated with the type II curve lies at $k = 0$, corresponding to $p = p_F$.

(iv) If $k \leq Q(\gamma)$, $p(-k) = 2p_F - p(k)$ and $\epsilon(-k) = \epsilon(k)$, hence $\epsilon^{II}(p) = \epsilon^{II}(2p_F - p)$, generalizing to finite interactions the symmetry found in the Tonks-Girardeau regime.

(v) The type I curve $\epsilon^I(p)$ repeats itself, starting from the umklapp point, shifted by $2p_F$ in $p$. Thus, what is usually considered as a continuation of the type II branch can also be thought as a shifted replica of the type I branch.

(vi) Close to the origin, $\epsilon^I(p) = -\epsilon^{II}(-p)$. This can be proven using the following sequence of equalities based on the previous properties:

$$\epsilon^I(p) = \epsilon^I(p + 2p_F) = -\epsilon^{II}(p + 2p_F) = -\epsilon^{II}(2p_F - (-p)) = -\epsilon^{II}(-p). \tag{45}$$

These properties will reveal most useful in the analysis of the spectra, they also provide stringent tests for numerical solutions. With the expansion method used before, we can obtain the type II curve explicitly, with excellent accuracy provided $\alpha > 2$. As far as the type I curve is concerned, however, we are not only limited by $\alpha > 2$, but also by the fact that our approximate expressions for $g(z;\alpha)$ and $f(z;\alpha)$ are valid only if $|z - y| \leq \alpha, \forall y \in [-1, 1]$, thus adding the validity condition, $|k|/Q(\alpha) \leq \alpha - 1$. The latter is not very constraining as long as $\alpha \gg 1$, but for $\alpha \simeq 2$ the best validity range we can get is very narrow around $p = 0$. To bypass this

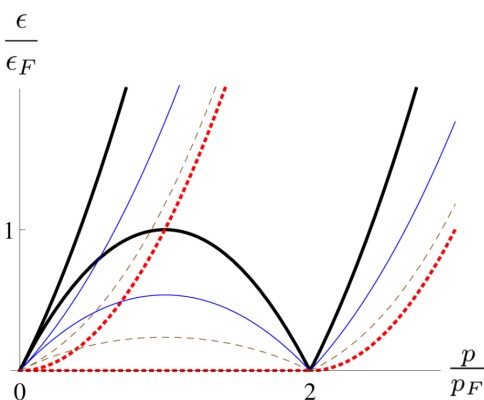

Figure 9: Type I and type II spectra for several values of the interaction strength, from the non-interacting Bose gas (red, dashed) [122] to the Tonks-Girardeau regime (black, solid, thick) with intermediate values $\alpha = 0.6$ (brown, dashed) and $\alpha = 2$ (blue, solid).

problem, Ristivojevic used an iteration method to evaluate $g$ and $f$ [1]. However, in practice this method is applicable only for large interactions since it allows to recover only the first few terms of the exact $1/\alpha$ expansion of $\epsilon(k;\alpha)$ and $p(k;\alpha)$ (to order 2 in [1]). Moreover, the obtained expressions are not of polynomial type, it is then a huge challenge to substitute the variable $k$ to express $\epsilon(p)$ explicitly, and one has to use approximate expressions at high and small momenta.

In the regime $|z| > 1$, we first compute the M-th order mean approximant for the function $g$, defined in Eq. (12),

$$g_m(z > 1; \alpha, M) = \frac{1}{2\pi} + \frac{\alpha}{\pi} \int_{-1}^{1} dy \, \frac{g_m(y;\alpha,M)}{\alpha^2 + (y-z)^2}, \tag{46}$$

which then allows us to obtain the type I spectrum with excellent accuracy for all values of $p$ from Eqs. (42) and (43). Both excitation spectra are shown in Fig. 9 for several values of $\gamma$. We note that the area below the type II spectrum, vanishing in the noninteracting Bose gas, is an increasing function of $\gamma$.

At small momenta, in the general case, due to the analytical properties of both $g$ and $f$, for all values of $\gamma$ the type I curve can be expressed as a series in $p$ [123, 124],

$$\epsilon^I(p;\gamma) = v_s(\gamma)p + \frac{p^2}{2m^*(\gamma)} + \frac{\lambda^*(\gamma)}{6}p^3 + \dots. \tag{47}$$

In the Tonks-Girardeau regime, as follows from Eq. (40), one has $v_s = v_F = \hbar k_F/m$, $m^* = m$, and all other coefficients vanish. At finite interaction strength, the parameters $v_s$ and $m^*$ can be seen as a renormalized Fermi velocity and mass respectively. Linear Luttinger liquid theory predicts that $v_s$ is the sound velocity associated with bosonic modes at very low $p$ [18]. To all accessed orders in $1/\gamma$, we have checked that our expression agrees with the exact thermodynamic equality [4]

$$v_s(\gamma) = \frac{v_F}{\pi} \left[ 3e(\gamma) - 2\gamma \frac{de}{d\gamma}(\gamma) + \frac{1}{2}\gamma^2 \frac{d^2e}{d\gamma^2}(\gamma) \right]^{1/2}. \tag{48}$$

Analytical expansions for the sound velocity are already known at large and small interaction strengths. The first- and second-order corrections to the Tonks-Girardeau regime in $1/\gamma$ are given in [25], they are calculated to fourth order in [87] and up to eighth order in [1].

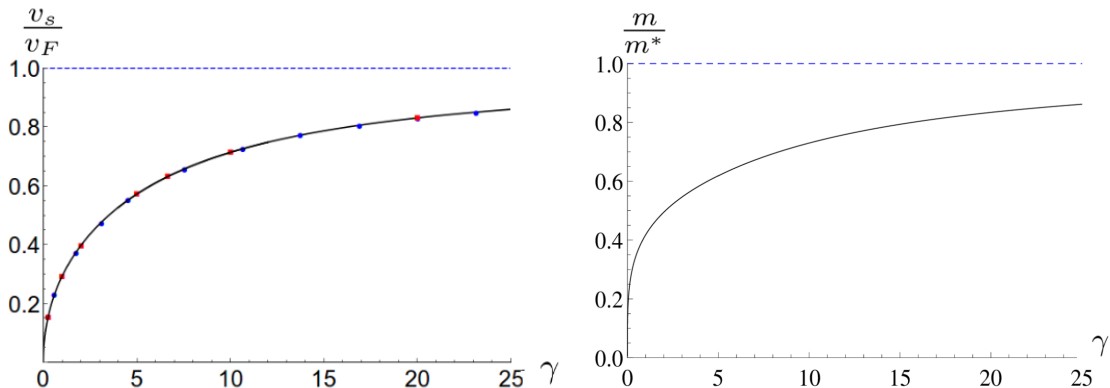

Figure 10: Left panel: dimensionless sound velocity $v_s/v_F$, where $v_F$ is the Fermi velocity, as a function of the dimensionless Lieb parameter $\gamma$, numerical (blue dots), numerical from literature [25, 34] (red squares), and our result (black, solid). The Tonks-Girardeau limit is indicated in dashed blue line in both panels. Right panel: dimensionless inverse renormalized mass $m/m^*$ obtained with Eqs. (50), (48), (23) and (10), as a function of the dimensionless Lieb parameter $\gamma$ (black).

In the weakly-interacting regime, expansions are found in [25] and [94]. Our expressions Eqs. (23, 24) for the dimensionless energy density allow us to considerably increase the accuracy compared to previous works after easy algebra. Interestingly, one does not need to compute the ground-state energy $e(\gamma)$ to find $v_s(\gamma)$. It is sufficient to know the $g$ function at $z = 1$ in the first Lieb equation due to the useful equality [32]

$$\frac{v_s(\gamma)}{v_F} = \frac{1}{[2\pi g(1; \alpha(\gamma))]^2}. \tag{49}$$

Reciprocally, knowing $v_s$ yields an excellent accuracy test for the $g$ function, since it allows to check its value at the border of the interval $[-1, 1]$ where it is the most difficult to obtain. Here we use both approaches to find the sound velocity over the whole range of interactions with excellent accuracy. Our results are shown in the left panel of Fig. 10. One can see that $v_s \to_{\gamma \to 0} 0$, thus we shall find $g(z; \alpha) \to_{z \to 1, \alpha \to 0} +\infty$, which shows that the method of polynomial expansion must fail at too low interaction strengths, as expected due to the presence of the singularity. Moreover, this argument automatically discards the approximate expression given in Eq. (7) close to the boundaries in $z$.

Linear Luttinger liquid theory assumes that, for all values of $\gamma$, $\epsilon^{I/II}(p) \simeq_{p/p_F \ll 1} v_s p$ and $\epsilon^{II}(p) \simeq_{|p-2p_F|/p_F \ll 1} v_s |p - 2p_F|$. This strictly linear spectrum is however a low-energy approximation, and nonlinearities cannot be neglected if one wants to deal with higher energies. Here, we provide the first quantitative study of the regime of validity of the linear approximation at finite interaction. We denote by $\Delta p$ and $\Delta \epsilon$ respectively the half-width of momentum, and maximum energy range around the umklapp point such that the linearized spectrum $\epsilon^{II} = v_s |p - 2p_F|$ is exact up to 10 percent. These quantities should be considered as upper bounds of validity for dynamical observables such as the dynamic structure factor [31]. Our results for $\Delta p$ and $\Delta \epsilon$ are shown in Fig. 11. One sees that Luttinger liquid theory works well at large interaction strengths. However, its range of validity decreases when interactions are decreased from the Tonks-Girardeau regime.

Including the quadratic term and neglecting higher-order ones is actually a complete change of paradigm, from massless bosonic to massive fermionic excitations at low energy, at the basis of the Imambekov-Glazman theory of nonlinear Luttinger liquids [125, 126]. In this approach $m^*$ is interpreted as an effective mass, whose general expression is [1, 127]

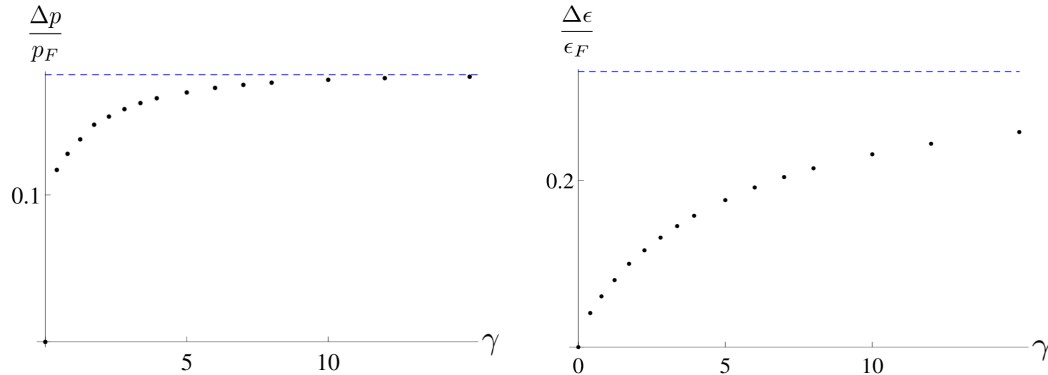

Figure 11: Upper bounds for the validity range in dimensionless momentum (left panel) and dimensionless energy (right panel) around the umklapp point $p = 2p_F, \epsilon = 0$ of the Luttinger liquid prediction for the excitation spectrum $\epsilon^{II}$, as functions of the dimensionless interaction strength $\gamma$. Dots represent the numerical estimate at finite interaction strength, the blue dashed curve is the exact result in the Tonks-Girardeau regime $\Delta \epsilon^{TG}/\epsilon_F \simeq 0.330581$ and $\Delta p^{TG}/p_F \simeq 0.18182$

$$\frac{m}{m^*} = \left( 1 - \gamma \frac{d}{d\gamma} \right) \sqrt{\frac{v_s}{v_F}}. \tag{50}$$

We have verified that the effective mass and the sound velocity obtained from the excitation spectrum satisfy Eq. (50) to all accessed orders in $1/\gamma$. Our results for the effective mass as obtained from a combination of the weak-coupling expression Eq. (10), the conjecture (23), and the use of Eqs. (48) and (50), is shown in the right panel of Fig. 10. We notice that the inverse effective mass vanishes for $\gamma \to 0$. This is also predicted by the Bogoliubov theory, where the small-$p$ expansion of the excitation dispersion reads $\epsilon^I_{Bog}(p) = v_s|p| + |p|^3/(8\sqrt{g_{1D}n_0})$. Hence, the non-vanishing inverse effective mass is a beyond-mean field effect.

For the type II spectrum, the properties (i)-(v) that we have detailed above suggest another type of expansion, also used in [128]:

$$\epsilon^{II}(p;\gamma) = \frac{1}{2m} \left[ f_1(\gamma)p(2p_F - p) + f_2(\gamma)\frac{p^2(2p_F - p)^2}{p_F^2} + \dots \right]. \tag{51}$$

Using the property (vi), on the other hand, allows us to write

$$\epsilon^{II}(p;\gamma) = v_s(\gamma)p - \frac{p^2}{2m^*(\gamma)} + \dots \tag{52}$$

Equating both expressions to order $p^2$, one finds that

$$f_1(\gamma) = \frac{v_s(\gamma)}{v_F}. \tag{53}$$

A similar result was recently inferred from Monte-Carlo treatment of 1D $^4$He and proved by Bethe Ansatz applied to the hard-rods model in [129, 130]. By the same approach we then show that

$$f_2(\gamma) = \frac{1}{4} \left( \frac{v_s(\gamma)}{v_F} - \frac{m}{m^*(\gamma)} \right), \tag{54}$$

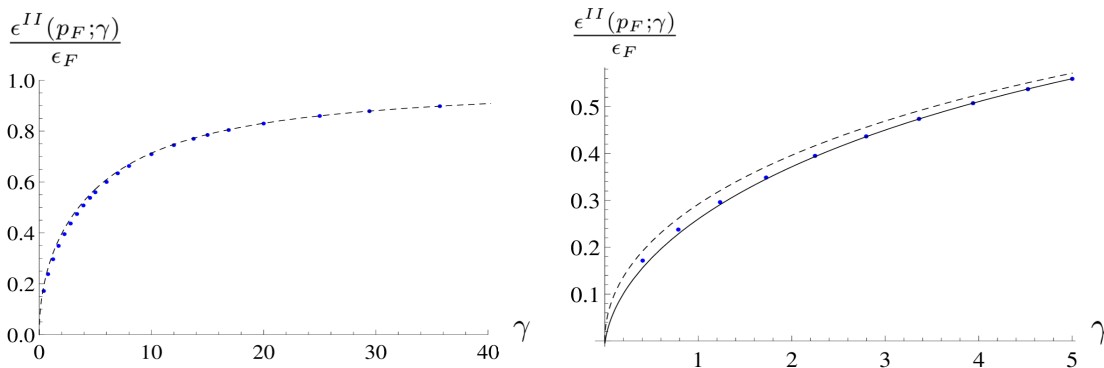

Figure 12: Maximum of the type II spectrum, $\epsilon^{II}(p_F; \gamma)$, in units of the Fermi energy $\epsilon_F$, as a function of the dimensionless interaction strength $\gamma$. The left panel shows the first order approximation in Eq. (51) taking $f_2 = 0$ (dashed), compared to exact numerics (blue dots). The right panel shows a zoom close to the origin and the second order approximation (solid, black). The agreement is significantly improved when using this correction.

which is one of our main new results.

Systematic suppression of the variable $k$ and higher-order expansions suggest that, at large enough values of $\gamma$ at least, higher-order terms in expansion (51) can be neglected. Figure 12 shows the value of the Lieb-II excitation spectrum at its local maximum value $\epsilon^{II}(p_F)$, as obtained from a numerical calculation as well as the expansion (51). We find that the result to order one is satisfying at large $\gamma$, but the second order correction significantly improves the result at intermediate values of the Lieb parameter. Our numerical calculations show that third and higher order corrections are negligeable in a wide range of strong interactions. Overall, we dispose of very accurate analytical predictions for the excitation branches, for $\gamma \in [1, +\infty[$.

# 4 Conclusions and outlook

In conclusion, first we have solved with high accuracy the set of Bethe-Ansatz equations Eqs. (4), (5) and (6) established by Lieb and Liniger for the ground state of a 1D gas of point-like bosons with contact repulsive interactions in the thermodynamic limit, thus obtaining the distribution of pseudo-momenta, the average energy per particle and all related quantities. Our main result in this part consists of two simple analytical expressions which describe to a good accuracy the ground state energy, namely, a weak coupling expansion, Eq. (10), valid for interaction strengths $\gamma \lesssim 15$, and a strong-coupling expansion to order 20, whose coefficients are given numerically in Eq. (104), valid for interaction strengths $\gamma \gtrsim 6$. Their combination spans the whole range of coupling constants, thus providing an alternative to the use of tabulated values and an opportunity to accurately benchmark numerical methods [131, 132].

More importantly, by a careful analysis of the strong-coupling expansion, we have found that the density of pseudo-momenta displays a peculiar structure, partially identified in Eqs. (14) to (20). We have also pointed out that doing the average of two consecutive even orders in the strong-coupling expansion dramatically increases the accuracy. The average between orders 18 and 20 in the inverse coupling $1/\alpha$ is very accurate for coupling constants as low as $\gamma \simeq 4.5$.

We have also proposed a conjecture for the ground-state energy valid at all interaction strengths, Eqs. (23) and (24), stating that the strong-coupling expansion of the dimensionless

ground-state energy resums through the following structure:

$$e(\gamma) = \gamma^2 \sum_{n=0}^{+\infty} \frac{\pi^{2(n+1)} P_n(\gamma)}{(2+\gamma)^{3n+2}}, \tag{55}$$

where $P_n$ is a $max(n-1,0)$-degree polynomial with rational coefficients, that are of alternate signs. This is a further step towards the exact closed-form solution of the model.

Then, we have studied the two branches of the excitation spectrum and found a new expression in terms of the sound velocity and effective mass. Both quantities can be obtained from the ground-state energy through thermodynamic relations, thus further showing the importance of an accurate knowledge of the ground state properties. In particular, we have identified the structure of the type-II branch in Eq. (51) as

$$\epsilon^{II}(p;\gamma) = \frac{1}{2m} \sum_{n=1}^{+\infty} f_n(\gamma) \frac{p^n (2p_F - p)^n}{p_F^{2(n-1)}}, \tag{56}$$

and identified $f_1$ and $f_2$ in Eqs. (53) and (54). Keeping only these two terms yields a very good approximation to the exact result for all interaction strengths. In turn, we have identified the validity range of Luttinger-liquid theory in the momentum-energy space. It works best in the Tonks-Girardeau regime and the range of validity decreases monotonically when the coupling constant is decreased.

A first natural generalization of our study concerns finite-temperature effects. The correlation functions have already been studied extensively, $g_2$ is known non-locally and at finite temperature [133–136], and the three- and more particle correlations too in certain regimes [137, 138], yet full-analytical explicit expressions are still limited to small ranges of temperature and interaction strength. Our results for the density of pseudo-momenta could be taken as a first numerical input in Monte-Carlo programs to solve the Bethe Ansatz equations at low temperature [31]. The guessing strategy might help at small temperatures where a Sommerfeld expansions can be used, but it is not obvious whether or not it can be extended to arbitrary temperatures, which is a complicated open problem.

We shall also mention recent works on correlation functions that do not tackle the Bethe Ansatz equations of the LL model directly. In [139,140], an appropriate nonrelativistic limit of the sinh-Gordon model is taken to find the form factors and thus evaluate the two- and three-body correlation functions at finite temperature and number of bosons for the LL model, also valid out of equilibrium. A complete resummation of the series involved was later performed in [141], and yields exact and compact integral equations satisfied by the correlations. Their solution can be seen as an improved $1/\gamma$ expansion. This approach, based on the 'LeClair-Mussardo formalism' [142], provides an independent way to check our conjectures by systematic comparison, and an alternative to Eq. (31). An other route, based on a continuum limit of the XXZ model yielding the LL model [143, 144], was taken in [145] to express correlations as multiple integrals that reduce to simple ones and in particular to compute $g_4$ at finite temperature. Further work in this direction [146] allowed the same author to show that the LeClair-Mussardo formalism can be deduced (and even generalized) from Bethe Ansatz, so that one actually does not need to consider the sinh-Gordon model. This very important result forsees a deep but yet not well understood link between relativistic Quantum Field Theories and the Algebraic Bethe Ansatz.

The harmonic trap used in most of current experiments would destroy the integrability by breaking translational invariance, but its effect on correlation functions can be studied numerically [147, 148] or within the local density approximation (LDA) [149]. In this respect, our improved results for the homogeneous gas can be used to increase the accuracy of theoretical predictions within LDA [150], and for comparison with exact results [151] to test the validity

of the LDA [152]. More generally, the effect of any integrability-breaking additional term in the Hamiltonian, if it is weak enough, can be evaluated within perturbation theory [153], yet Bethe Ansatz techniques are not versatile enough to tackle them in full generality, so one still needs to rely on numerical methods.

The very accurate expressions we have obtained for the excitation spectra may reveal useful to better understand the link between type II excitations and quantum dark solitons. They have been mostly studied in the weakly-interacting regime so far [154–160], but they may be a general feature of the model [161–164]. Excitation spectra also yield the exponents governing the shape of the dynamical structure factor near the edges in the framework of 'beyond Luttinger liquid' theory [41, 126].

All our techniques can be adapted easily to the metastable gaseous branch of bosons with attractive interactions [165–167], to study the super Tonks-Girardeau behavior of the model with increased accuracy. Extensions of the current method may be used to study the Yang-Gaudin model of spinful 1D fermions, which has attracted much attention recently [168–174]. Furthermore, since the Lieb-Liniger model can be seen as the special case of infinitely many different spin values [175], it allows to check the consistency with the general case, which is far less well understood, and to make approximate predictions at the highest experimentally relevant values of the number of spin components.

*Note added: soon after the first version of our article appeared on the arXiv, considerable progress in the evaluation of the ground-state energy was made by Sylvain Prolhac. In particular, he has numerically evaluated several new terms in Eq. (9) with outstanding accuracy [101]. Our equation (104) is in perfect agreement with his results, as well as the polynomials in our conjecture Eq. (24), and the highest-degree monomial has the form we have inferred even at higher orders [102].*

## Acknowledgements

We thank Fabio Franchini, Maxim Olshanii and Zoran Ristivojevic for useful discussions, and Martón Kormos for insightful comments regarding the first version. We thank Sylvain Prolhac for invaluable comments and discussions.

We acknowledge financial support from ANR project Mathostaq (ANR-13-JS01-0005-01), ANR project SuperRing (ANR-15-CE30-0012-02) and from Institut Universitaire de France.

## A   Link between the Lieb equation and the capacitance of the circular plate capacitor

In this appendix we illustrate the exact mapping between the Lieb-Liniger model discussed in the main text and a problem of classical physics. Both have beneficiated from each other and limiting cases can be understood in different ways according to the context.

Capacitors are emblematic systems in electrostatics undergraduate courses. On the example of the parallel plate ideal capacitor, one can introduce various concepts such as symmetries of fields or Gauss law, and compute the capacitance in a few lines from basic principles, assuming that the plates are infinite (or at contact). To go beyond this approximation, geometry must be taken into account to include edge effects, as was realized by Clausius, Maxwell and Kirchhoff in pioneering tentatives to include them [176–178]. Actually, the exact capacitance of a circular coaxial plate capacitor with a free space gap as dielectrics, as a function of the aspect ratio of the cavity $\alpha = d/R$, where $d$ is the distance between the plates and $R$ their

radius, reads [179]

$$C(\alpha,\lambda) = 2\epsilon_0 R \int_{-1}^{1} dz\, g(z;\alpha,\lambda), \tag{57}$$

where $\epsilon_0$ is the permittivity of vacuum, $\lambda = \pm 1$ in the case of equal (respectively opposite) disc charge or potential and $g$ is the solution of the Love equation [180, 181]

$$g(z;\alpha,\lambda) = 1 + \frac{\lambda\alpha}{\pi} \int_{-1}^{1} dy\, \frac{g(y;\alpha,\lambda)}{\alpha^2 + (y-z)^2}, \quad -1 \le z \le 1. \tag{58}$$

This equation turns out to be the Lieb equation Eq. (4) when $\lambda = 1$, as first noticed by Gaudin [182]. In this exact mapping, however, the relevant physical quantities are different and are obtained at different steps of the resolution. In what follows, we consider the case of equally charged discs and do not write the index $\lambda$ anymore.

At small $\alpha$, i.e. at small gaps, using Eq. (7) one finds

$$C(\alpha) \simeq_{\alpha \ll 1} \frac{\pi\epsilon_0 R}{\alpha} = \frac{\epsilon_0 A}{d}, \tag{59}$$

where $A$ is the area of a plate, as directly found in the contact approximation. On the other hand, if the plates are taken apart from each other up to infinity (this corresponds to the Tonks-Girardeau regime in the Lieb-Liniger model), one finds $g(z;+\infty) = 1$ and thus

$$C(\alpha) =_{\alpha \to +\infty} 4\epsilon_0 R. \tag{60}$$

This result can be understood as follows. At large distance, the two plates do not feel each others and can be considered as being in series. The capacitance of one plate is $8\epsilon_0 R$, and the additivity of inverse capacitances in series yields the awaited result. At intermediate distances, one qualitatively expects that the exact capacitance is larger than the value found in the contact approximation, due to the fringing electric field outside the cavity delimited by the two plates. The contact approximation shall thus yield a lower bound for any value of $\alpha$.

Main results and conjectures in the small $\alpha$ regime [92, 183–186] are summarized in [187] and all encompassed in the most general form

$$\mathscr{C}(\epsilon) = \frac{1}{8\epsilon} + \frac{1}{4\pi}\log\left(\frac{1}{\epsilon}\right) + \frac{\log(8\pi)-1}{4\pi} + \frac{1}{8\pi^2}\epsilon\log^2(\epsilon) + \sum_{i=1}^{+\infty}\epsilon^i\sum_{j=0}^{2i}c_{ij}\log^j(\epsilon), \tag{61}$$

where notations are $\epsilon = \frac{\alpha}{2}$, and $\mathscr{C} = C/(4\pi\epsilon_0 R)$ is the geometrical capacitance. It is known that $c_{12} = 0$ [187]. In the same reference, a link with differential geometry is found and discussed but lies beyond the scope of our work. Moreover,

$$\mathscr{C}_{\le}(\epsilon) = \frac{1}{8\epsilon} + \frac{1}{4\pi}\log\left(\frac{1}{\epsilon}\right) + \frac{\log(4)-\frac{1}{2}}{4\pi} \tag{62}$$

is a sharp lower bound as shown in [184].

At large $\alpha$, i.e. for distant plates, many different techniques have been considered over the years. Historically, Love used the iterated kernel method. Injecting the right-hand side of Eq. (58) into itself and iterating, one can express the solution as a Neumann series [180]

$$g(z;\alpha,\lambda) = 1 + \sum_{n=1}^{+\infty}\lambda^n\int_{-1}^{1}K_n(y-z)dy \equiv \sum_{n=0}^{+\infty}\lambda^n g_n^I(z;\alpha) \tag{63}$$

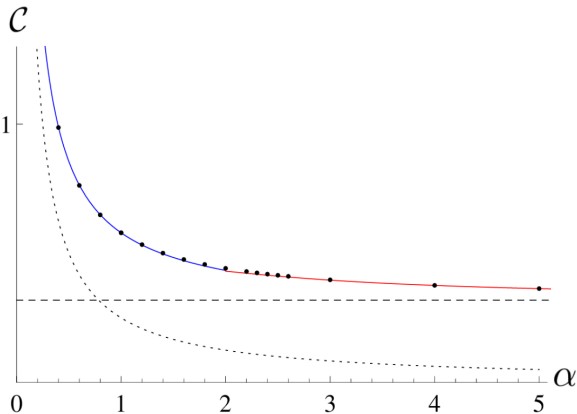

Figure 13: Geometric dimensionless capacitance $\mathscr{C}$ of the paralell plate capacitor as a function of its dimensionless aspect ratio $\alpha$. Results at infinite gap (dotted) and in the contact approximation (dotted) are actually rather crude compared to the more sophisticated approximate expressions from Eq. (61) for $\alpha < 2$ and Eq. (66) for $\alpha > 2$ (solid blue and red respectively), compared to numerical solution of Eqs. (57,58) (black dots).

with $\quad K_1(y-z;\alpha) = \frac{\alpha}{\pi}\frac{1}{\alpha^2+(y-z)^2}\quad$ the kernel of Love's equation, and $K_{n+1}(y-z;\alpha) \equiv \int_{-1}^{1} dx\, K_1(y-x;\alpha)K_n(x-z;\alpha)$ the iterated kernels. It follows easily that for repulsive plates, $g(z;\alpha,+1) > 1$, yielding a global lower bound in agreement with the physical discussion above. One also finds that $g(z;\alpha,-1) < 1$. Approximate solutions are then obtained by truncation to a given order. One easily finds that

$$g_1^I(z;\alpha) = \frac{1}{\pi}\left[\arctan\left(\frac{1-z}{\alpha}\right) + \arctan\left(\frac{1+z}{\alpha}\right)\right], \tag{64}$$

yielding

$$C_1^I(\alpha) = 2\epsilon_0 R\left[4\arctan\left(\frac{2}{\alpha}\right) + \alpha\log\left(\frac{\alpha^2}{\alpha^2+4}\right)\right], \tag{65}$$

where $C(\alpha,\lambda) = \sum_{n=0}^{+\infty}\lambda^n C_n^I(\alpha)$. However, higher orders are cumbersome to evaluate, which is a strong limitation of this method. Among alternative ways to tackle the problem, we mention Fourier series expansion [179, 188], and those based on orthogonal polynomials [191], that allowed to find the exact expansion of the capacitance at order 9 in $1/\alpha$ in [189] for identical plates, anticipating [1].

In Fig. (13), we show several approximations of the geometric capacitance as a function of the aspect ratio. In particular, based on an analytical asymptotic expansion, we propose a simple approximation in the large gap regime

$$\mathscr{C}(\alpha) \simeq_{\alpha\gg 1} \frac{1}{\pi}\frac{1}{1-2/(\pi\alpha)} - \frac{4}{3\pi^2\alpha^3}\frac{1}{(1-2/(\pi\alpha))^2}. \tag{66}$$

## B   An accuracy test of solutions to the Lieb equation

In this Appendix we introduce tools to study the accuracy of a given approximate solution of Eq. (4). We define a local error functional by

$$\epsilon[g;\alpha](z) = g(z;\alpha) - \frac{1}{2\pi}\int_{-1}^{1} dy\,\frac{2\alpha g(y;\alpha)}{\alpha^2+(y-z)^2} - \frac{1}{2\pi}, \tag{67}$$

and the corresponding global error functional

$$E[g;\alpha] = \int_{-1}^{1} dz |\epsilon[g;\alpha](z)|. \tag{68}$$

If a proposed solution $g(z;\alpha)$ is exact for a given fixed parameter $\alpha$, then trivially $\epsilon[g;\alpha]$ as a function of $z$ is uniformly zero. A sufficient condition for an approximate solution $g$ to be more accurate than an other solution $\tilde{g}$ is that $|\epsilon[g;\alpha]| \leq |\epsilon[\tilde{g};\alpha]|$ for all $z$ in $[-1,1]$. The global error functional yields a good accuracy criterion, by requiring that it is lower than a threshold. Both quantities can be used for numerical as well as analytical purposes. We used them to check that the correction in Eq. (8) improves locally the accuracy with respect to Eq. (7) close to $z = 0$ whenever $\alpha < 1$. However, close to $|z| = 1$ we find that Eq. (8) is not necessarily more accurate.

We also propose a criterion specifically designed to deal with the case $\alpha = 1$. Since $g$ is analytic in $z$ and even, using the property [192] $\int_0^1 \frac{x^\mu}{1+x^2} dx = \frac{1}{2}\beta(\frac{\mu+1}{2})$, where $\beta$ is the beta function, defined as $\beta(x) \equiv \frac{1}{2}\left[\psi\left(\frac{x+1}{2}\right) - \psi\left(\frac{x}{2}\right)\right]$, with $\psi$ the logarithmic derivative of the Euler $\Gamma$ function, also known as the digamma function, we naturally define an error functional by

$$Err[g] = \frac{1}{2}g(0;1) - \frac{1}{\pi}\sum_{k=1}^{+\infty} \frac{g^{(2k)}(0;1)}{(2k)!}\beta\left(\frac{2k+1}{2}\right) - \frac{1}{2\pi}. \tag{69}$$

For instance, in [180] the following expression was proposed:

$$g(z;1) = 0.305450 - 0.049611z^2 + 0.002495z^4 + 0.0031325z^6 - 0.000059z^8. \tag{70}$$

Then $|Err[g]| \simeq 0.0032 \ll g(0,1)$. Our numerical result is very close to that function. Fitting it by an eighth-degree polynomial, we find exactly the same value for the error up to $4^{th}$ digit. This allows us to check once more the accuracy of our numerical algorithm.

## C  A method to solve the Lieb equation for $\alpha > 2$

In this Appendix we give our derivation of Ristivojevic's method [1] to systematically find approximate solutions to Eq. (4) in the strongly-interacting regime. First, we recall some qualitative features of the function of interest, $g(z;\alpha)$. In [2,180] it was shown that at fixed $\alpha$, $g$ as a function of $z$ is positive, bounded, unique and even. Moreover, it is analytic provided $\alpha > 0$.

Since $g$ is an analytic function of $z$, due to a theorem from Weierstrass, in $[-1,1]$ and at fixed $\alpha$ it can be written as $g(z;\alpha) = \sum_{n=0}^{+\infty} a_n(\alpha)Q_n(z)$, where $a_n$ are unknown regular functions and $Q_n$ are polynomials of degree $n$.

To solve the system Eqs. (4), (5) and (6), one only needs the values of $g$ for $z \in [-1,1]$. Thus, a good basis for the $Q_n$'s is provided by the Legendre polynomials $P_n(X) \equiv \frac{(-1)^n}{2^n n!}\left(\frac{d}{dX}\right)^n [(1-X^2)^n]$, which form a complete orthogonal set in this range. Furthermore, Legendre polynomials of degree $n$ consist of sums of monomials of the same parity as $n$, so that, since $g$ is even in $z$,

$$g(z;\alpha) = \sum_{n=0}^{+\infty} a_{2n}(\alpha)P_{2n}(z). \tag{71}$$

If one restricts to $\alpha > 2$, since $y, z \in [-1, 1]$, the Lorentzian kernel in Eq. (4) can be expanded as:

$$\frac{1}{\pi} \frac{\alpha}{\alpha^2 + (y-z)^2} = \frac{1}{\pi} \sum_{k=0}^{+\infty} \frac{(-1)^k}{\alpha^{2k+1}} \sum_{j=0}^{2k} \binom{2k}{j} y^j (-1)^j z^{2k-j}. \tag{72}$$

Thus, the combination of Eqs. (4), (71) and (72) yields

$$\sum_{n=0}^{+\infty} a_{2n}(\alpha) \left[ P_{2n}(z) - \frac{1}{\pi} \sum_{k=0}^{+\infty} \frac{(-1)^k}{\alpha^{2k+1}} \sum_{j=0}^{2k} \binom{2k}{j} (-1)^j z^{2k-j} \int_{-1}^{1} dy \, y^j P_{2n}(y) \right] = \frac{1}{2\pi}. \tag{73}$$

Following [1], we introduce the notation $F_{2n}^j \equiv \int_{-1}^{1} dy \, y^j P_{2n}(y)$. Due to the parity, $F_{2n}^j \neq 0$ if and only if $j$ is even. An additional condition is that $j \geq n$ [192]. Taking it into account and renaming mute parameters ($k \leftrightarrow n$) yields

$$\sum_{n=0}^{+\infty} \left[ a_{2n}(\alpha) P_{2n}(z) - \frac{1}{\pi} \sum_{j=0}^{n} \sum_{k=0}^{j} \frac{(-1)^n}{\alpha^{2n+1}} a_{2k}(\alpha) \binom{2n}{2j} z^{2(n-j)} F_{2k}^{2j} \right] = \frac{1}{2\pi}. \tag{74}$$

To go further, we use the property of orthogonality and normalization of Legendre polynomials: $\int_{-1}^{1} P_i(z) P_j(z) dz \neq 0$ if and only if $i = j$ since $i$ and $j$ are even, and $\int_{-1}^{1} dz P_j(z)^2 = \frac{2}{2j+1}$. Doing $\int_{-1}^{1} dz P_{2m}(z) \times$Eq. (74) yields:

$$\sum_{n=0}^{+\infty} \left[ a_{2n}(\alpha) \delta_{m,n} \frac{2}{4m+1} - \frac{1}{\pi} \sum_{j=0}^{n} \sum_{k=0}^{j} \frac{(-1)^n}{\alpha^{2n+1}} a_{2k}(\alpha) \binom{2n}{2j} F_{2k}^{2j} F_{2m}^{2(n-j)} \right] = \frac{1}{2\pi} F_{2m}^0. \tag{75}$$

or after $n - m \to n$:

$$\frac{2a_{2m}(\alpha)}{4m+1} - \frac{1}{\pi} \sum_{n=0}^{+\infty} \sum_{j=0}^{n} \sum_{k=0}^{j} \frac{(-1)^{n+m}}{\alpha^{2(n+m)+1}} a_{2k}(\alpha) \binom{2(n+m)}{2j} F_{2k}^{2j} F_{2m}^{2(n+m-j)} = \frac{1}{2\pi} F_0^0 \delta_{m,0}. \tag{76}$$

Then, from equation 7.231.1 of [192] and after a few lines of algebra,

$$F_{2m}^{2l} = \frac{2^{2m+1}(2l)!(l+m)!}{(2l+2m+1)!(l-m)!}. \tag{77}$$

Inserting Eq. (77) into Eq. (76) yields after a few simplifications:

$$\frac{2a_{2m}(\alpha)}{4m+1} - \frac{1}{\pi} \sum_{n=0}^{+\infty} \sum_{j=0}^{n} \sum_{k=0}^{j} \frac{(-1)^{n+m}}{\alpha^{2(n+m)+1}} C_{m,n,j,k} a_{2k}(\alpha) = \frac{1}{\pi} \delta_{m,0} \tag{78}$$

where

$$C_{m,n,j,k} \equiv \frac{2^{2k+1}(j+k)!}{(2j+2k+1)!(j-k)!} \frac{2^{2m+1}(n+2m-j)!(2n+2m)!}{(2n+4m-2j+1)!(n-j)!}. \tag{79}$$

To make the system of equations finite, we cut off the series in $n$ at an integer value $M \geq 0$. The system Eq. (78) truncated at order $M$ can then be recast into a matrix form:

$$[A] \begin{bmatrix} a_0 \\ a_2 \\ \vdots \\ a_{2M} \end{bmatrix} = \begin{bmatrix} \frac{1}{\pi} \\ 0 \\ \vdots \\ 0 \end{bmatrix} \tag{80}$$

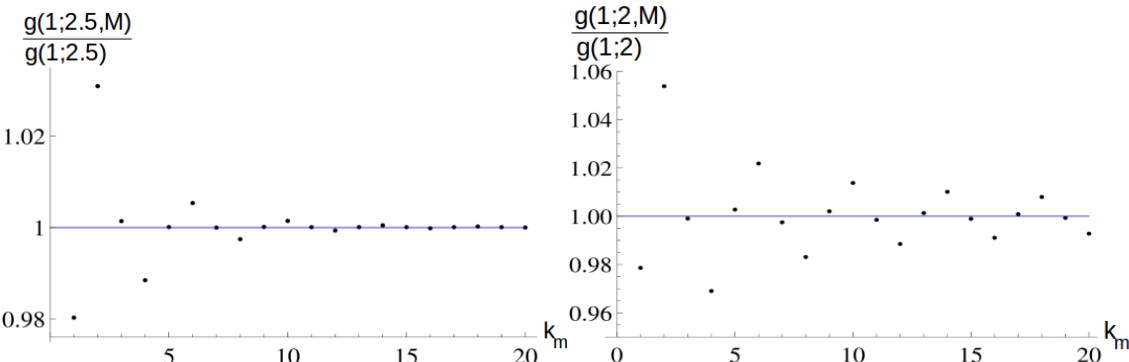

Figure 14: Dimensionless ratio $g(1; \alpha, M)/g(1; \alpha)$ of the approximate analytical solution and the numerically exact solution of Eq. (4) for several values of the maximal order of expansion $k_m$, at dimensionless parameters $\alpha = 2.5 \leftrightarrow \gamma = 6.0257$ (left panel) and $\alpha = 2 \leftrightarrow \gamma_c = 4.5268$ (right panel). The ratio 1 corresponds to the exact solution and is indicated by the blue line as a guide to the eye. When $\alpha$ is decreased, larger values of the parameter $M$, or equivalently $k_m (\leq 2M+2)$, are needed to reach a given accuracy. Quite remarkably, due to the oscillation of the analytics around the exact solution, odd-order terms are more accurate than even ones.

where $A$ is a $(M + 1) \times (M + 1)$ square matrix, which is inverted to find the set of coefficients $a_{2n}(\alpha)$. Actually, one only needs to compute $(A^{-1})_{i1}, \forall i \in \{1, \ldots, M+1\}$, then combined with Eq. (71) to obtain the final result at order $M$. For full consistency with higher orders, one shall expand the result in $1/\alpha$ and truncate it at order $2M+2$, while it goes to order $2M$ in $z$.

In order to check the accuracy of the analytical results, at fixed $\alpha$ we compare the various approximations we find at increasing $M$ with an accurate numerical solution obtained with a Monte-Carlo integration algorithm stopping when the condition on the global error functional $E[g; \alpha] < h$ is fulfilled, where $h$ is a threshold value fixed at $10^{-5}$ for $\alpha > 2$. We only need to compare the values at $z = 1$, which are the most difficult to attain analytically, to get an idea of the accuracy of the expansion. A systematic comparison at increasing values of $M$ is shown in Fig. 14. As explained in Appendix C, at fixed $M$ one obtains simultaneously the expansions to orders $k_m = 2M+1$ and $2M+2$ in $1/\alpha$. Since we look for fully analytical solutions we are limited to relatively low values of $M$. We solved the Lieb equation up to $M = 9$ and checked that our result agrees with the Supplementary Material of [1], where it is given to order $M = 3$. In particular, again we study the worst case for our method, $\alpha = 2$, where the convergence is the slowest with $M$ because this value lies on the border of the convergence domain. In [189] it was estimated that an expansion to order $k_m \simeq 40$ is needed to reach a 5-digit accuracy. We find that at $k_m = 20$ the relative error is still as large as $8/1000$. As a general fact, approximate solutions oscillate with $M$ around the exact solution, and odd orders are more accurate, as pointed out in the same reference.

We remark that, as a rule of thumb, the absolute value of the local error defined in Eq. (67) increases when $\alpha$ decreases and when $z$ increases. Thus, very high order asymptotic expansions may be required to get an accurate description up to $\alpha = 2$, which is the lowest attainable value in this approach.

# D   General method to solve the Lieb equation

The main drawback of the method to solve Eq. (4) exposed in Appendix C is its range of validity, limited to $\alpha > 2$. It can be further improved to circumvent this problem, as we show

here. Since $g$ as a function of $z$ is analytic and even, one can write

$$g(z,\alpha) = \sum_{n=0}^{+\infty} b_{2n}(\alpha) z^{2n}. \tag{81}$$

Here, we do not expand the kernel nor use orthogonal polynomials, but directly evaluate the integral

$$I(z,\alpha) \equiv \int_{-1}^{1} dy \, \frac{y^{2n}}{\alpha^2 + (y-z)^2}. \tag{82}$$

According to [193],

$$I(z,\alpha) = I_1(z,\alpha) + I_2(z,\alpha) + I_3(z,\alpha) \tag{83}$$

with

$$I_1(z,\alpha) = -nz^{2n-1} F\left(1-n, \frac{1}{2}-n; \frac{3}{2}; -\frac{\alpha^2}{z^2}\right) \ln\left(\frac{(1+z)^2 + \alpha^2}{(1-z)^2 + \alpha^2}\right), \tag{84}$$

$$I_2(z,\alpha) = \frac{z^{2n}}{\alpha} F\left(-n, \frac{1}{2}-n; \frac{1}{2}; -\frac{\alpha^2}{z^2}\right) \left[\arctan\left(\frac{1+z}{\alpha}\right) + \arctan\left(\frac{1-z}{\alpha}\right)\right], \tag{85}$$

and

$$I_3(z,\alpha) = 2 \sum_{m=0}^{n-1} \frac{z^{2m}}{2n-2m-1} (2m+1) F\left(-m, \frac{1}{2}-m : \frac{3}{2}; -\frac{\alpha^2}{z^2}\right). \tag{86}$$

where $F$ represents the Euler hypergeometric function, often denoted $_2F_1$.

To simplify these expressions, we systematically express the hypergeometric functions in terms of standard ones.

First, to simplify $I_1$ we use [192]

$$F\left(1, \frac{1}{2}; \frac{3}{2}; -\frac{\alpha^2}{z^2}\right) = \frac{z}{\alpha} \arctan\left(\frac{\alpha}{z}\right) \tag{87}$$

if $n = 0$, and if $n \neq 0$,

$$F\left(1-n, \frac{1}{2}-n; \frac{3}{2}; -\frac{\alpha^2}{z^2}\right) = \frac{\sin\left(2n\arctan\left(\frac{\alpha}{z}\right)\right)}{2n \sin\left(\arctan\left(\frac{\alpha}{z}\right)\right) \cos^{2n-1}\left(\arctan\left(\frac{\alpha}{z}\right)\right)}. \tag{88}$$

Basic trigonometry yields

$$\sin(\arctan(y)) = \frac{y}{\sqrt{1+y^2}} \tag{89}$$

and

$$\cos(\arctan(y)) = \frac{1}{\sqrt{1+y^2}}. \tag{90}$$

Moreover [192],

$$\sin(nx) = \sum_{k=0}^{\left[\frac{n-1}{2}\right]} (-1)^k \binom{n}{2k+1} \cos^{n-(2k+1)}(x) \sin^{2k+1}(x), \tag{91}$$

where $[x]$ represents the integer part of $x$. Combining these properties, we obtain

$$I_1(z,\alpha) = -\frac{1}{\alpha} \sum_{k=0}^{\left[\frac{2n-1}{2}\right]} \binom{2n}{2k+1} (-1)^k \alpha^{2k+1} z^{2n-(2k+1)} \frac{1}{2} \ln\left(\frac{(1+z)^2 + \alpha^2}{(1-z)^2 + \alpha^2}\right). \tag{92}$$

We proceed by evaluating $I_2$. According to [192],

$$F\left(-\frac{m}{2}, \frac{1}{2} - \frac{m}{2}; \frac{1}{2}; -\tan^2(x)\right) = \frac{\cos(mx)}{\cos^m(x)}. \tag{93}$$

Furthermore,

$$\cos(nx) = \sum_{k=0}^{I\left(\frac{n}{2}\right)} (-1)^k \binom{n}{2k} \cos^{n-2k}(x) \sin^{2k}(x), \tag{94}$$

thus

$$I_2(z,\alpha) = \frac{1}{\alpha}\left[\arctan\left(\frac{1+z}{\alpha}\right) + \arctan\left(\frac{1-z}{\alpha}\right)\right] \sum_{k=0}^{n} (-1)^k \binom{2n}{2k} (-1)^k \alpha^{2k} z^{2n-2k} \tag{95}$$

We finish by evaluating $I_3$. According to [192],

$$F(\alpha,\beta;\gamma;z) = (1-z)^{\gamma-\alpha-\beta} F(\gamma-\alpha, \gamma-\beta; \gamma; z) \tag{96}$$

and

$$F\left(\frac{n+2}{2}, \frac{n+1}{2}; \frac{3}{2}; -\tan^2(z)\right) = \frac{\sin(nz)\cos^{n+1}(z)}{n\sin(z)} \tag{97}$$

yielding

$$F\left(-m, \frac{1}{2} - m; \frac{3}{2}; -\frac{\alpha^2}{z^2}\right) = \left(1 + \frac{\alpha^2}{z^2}\right)^{2m+1} \frac{\sin\left((2m+1)\arctan\left(\frac{\alpha}{z}\right)\right)\cos^{2m+2}\left(\arctan\left(\frac{\alpha}{z}\right)\right)}{(2m+1)\sin\left(\arctan\left(\frac{\alpha}{z}\right)\right)}, \tag{98}$$

that simplifies into

$$F\left(-m, \frac{1}{2} - m; \frac{3}{2}; -\frac{\alpha^2}{z^2}\right) = \sum_{k=0}^{m} (-1)^k \binom{2m+1}{2k+1} \left(\frac{\alpha}{z}\right)^{2k} \frac{1}{2m+1} \tag{99}$$

hence

$$I_3(z,\alpha) = 2 \sum_{m=0}^{n-1} \sum_{k=0}^{m} \binom{2m+1}{2k+1} \frac{1}{2n-2m-1} (-1)^k \alpha^{2k} z^{2(m-k)}. \tag{100}$$

Eventually, combining all those results, one finds

$$\begin{aligned}
I(z,\alpha) = &-\frac{1}{\alpha} \sum_{k=0}^{\left[\frac{2n-1}{2}\right]} \binom{2n}{2k+1} (-1)^k \alpha^{2k+1} z^{2n-(2k+1)} \frac{1}{2} \ln\left(\frac{(1+z)^2 + \alpha^2}{(1-z)^2 + \alpha^2}\right) \\
&+ \frac{1}{\alpha}\left[\arctan\left(\frac{1+z}{\alpha}\right) + \arctan\left(\frac{1-z}{\alpha}\right)\right] \sum_{k=0}^{n} (-1)^k \binom{2n}{2k} (-1)^k \alpha^{2k} z^{2n-2k} \\
&+ 2 \sum_{m=0}^{n-1} \sum_{k=0}^{m} \binom{2m+1}{2k+1} \frac{1}{2n-2m-1} (-1)^k \alpha^{2k} z^{2(m-k)} \\
&\equiv \sum_{i=0}^{+\infty} d_{2i,n}(\alpha) z^{2i}.
\end{aligned} \tag{101}$$

The Lieb equation is thus recast into the form

$$\sum_{n=0}^{+\infty} b_{2n}(\alpha)\left[z^{2n} - \frac{\alpha}{\pi}\sum_{i=0}^{+\infty} d_{2i,n}(\alpha)z^{2i}\right] = \frac{1}{2\pi}. \tag{102}$$

One can truncate the infinite sum to order $M$ in a self-consistent way to obtain the following system of $M$ linear equations:

$$b_{2n;M}(\alpha) - \frac{\alpha}{\pi}\sum_{i=0}^{M} d_{2n,i} b_{2i;M}(\alpha) = \delta_{n,0}\frac{1}{2\pi}, \tag{103}$$

whose solution yields approximate expressions for a truncated polynomial expansion of $g$ in $z$. This algorithm is actually far more efficient than the less general one in their common range of validity, since $b_{2n}(\alpha)$ decreases at a very fast rate when $n$ increases, at fixed $\alpha$, and due to the hierarchy of coefficients. Nonetheless, below $\alpha = 2$ a larger number of terms is needed, and expressions become progressively longer at a very fast pace.

# E  Expansion of the dimensionless energy per particle in the strongly-interacting regime

Using the method of Appendix C up to $M = 9$ yields the expansion of $e(\gamma)$ to order 20 in $1/\gamma$. The expression is given here with numerical coefficients for the sake of compactness, and reads

$$
\begin{aligned}
e(\gamma) \simeq{}& 1. - \frac{4.}{\gamma} + \frac{12.}{\gamma^2} - \frac{10.9448}{\gamma^3} - \frac{130.552}{\gamma^4} + \frac{804.13}{\gamma^5} - \frac{910.345}{\gamma^6} - \frac{15423.8}{\gamma^7} + \frac{100559.}{\gamma^8} - \frac{67110.5}{\gamma^9} \\
&- \frac{2.64681*10^6}{\gamma^{10}} + \frac{1.55627*10^7}{\gamma^{11}} + \frac{4.69185*10^6}{\gamma^{12}} - \frac{5.35057*10^8}{\gamma^{13}} + \frac{2.6096*10^9}{\gamma^{14}} + \frac{4.84076*10^9}{\gamma^{15}} \\
&- \frac{1.16548*10^{11}}{\gamma^{16}} + \frac{4.35667*10^{11}}{\gamma^{17}} + \frac{1.93421*10^{12}}{\gamma^{18}} - \frac{2.60894*10^{13}}{\gamma^{19}} + \frac{6.51416*10^{13}}{\gamma^{20}} + O\left(\frac{1}{\gamma^{21}}\right).
\end{aligned}
\tag{104}
$$

It is illustrated in Fig. (15).

# F  A method to solve the second Lieb equation for $\alpha > 2$

To find the type I and type II dispersion relations, we need to solve Eq. (44) copied here for convenience:

$$f(z;\alpha) - \frac{1}{\pi}\int_{-1}^{1} dy \frac{\alpha}{\alpha^2 + (y-z)^2} f(y;\alpha) = z. \tag{105}$$

We noted that the same equation has been reported to occur in other physical contexts, such as hydrodynamics near a submerged disk [194, 195]. Adapting the techniques of Appendix C, using odd-degree Legendre polynomials because $f$ is an odd function of $z$, we write

$$f(z;\alpha) = \sum_{n=0}^{+\infty} a_{2n+1}(\alpha)P_{2n+1}(z), \tag{106}$$

and find

$$\frac{2a_{2m+1}(\alpha)}{4m+3} - \frac{1}{\pi}\sum_{n=0}^{M}\sum_{j=0}^{n}\sum_{k=0}^{j}(-1)^{n+m}\tilde{C}_{mnjk}\frac{a_{2k+1}(\alpha)}{\alpha^{2(n+m)+3}} \simeq \frac{2}{3}\delta_{m,0}, \tag{107}$$

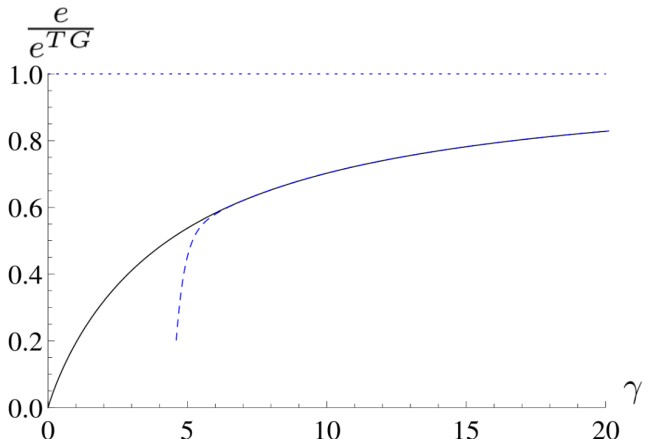

Figure 15: Ground state energy per particle $e$ in units of its value in the Tonks-Girardeau limit $e^{TG}$, as a function of the dimensionless interaction strength $\gamma$. Numerically exact result (black), compared to $20^{th}$-order expansion in the strongly-interacting regime (blue, dashed) as given by Eq. (104). The value in the Tonks-Girardeau limit (horizontal, blue, dotted) is also indicated on the figure.

with

$$\tilde{C}_{mnjk} \equiv \frac{2^{2(k+m+2)}(2m+2n+2)!(j+k+1)!(n+2m-j+1)!}{(2k+2j+3)!(j-k)!(2n+4m-2j+3)!(n-j)!}. \tag{108}$$

Equation (107) at order $M$ is rewritten into a matrix form:

$$\begin{bmatrix} B \end{bmatrix} \begin{bmatrix} a_1 \\ a_3 \\ \vdots \\ a_{2M+1} \end{bmatrix} = \begin{bmatrix} \frac{2}{3} \\ 0 \\ \vdots \\ 0 \end{bmatrix} \tag{109}$$

where $B$ is a $(M+1) \times (M+1)$ square matrix. Actually, one only needs to compute $B_{i1}^{-1}$, $\forall i \in \{1, \dots, M+1\}$.

For the same reasons as before, the expansion is valid for $\alpha > 2$, i.e. in the strongly-interacting regime. At fixed $\alpha$ it converges faster than $g$ to the exact value when $M$ is increased. Once again, from the expansions obtained one can guess patterns and write

$$f(z) = z + \sum_{m,n} J_{m,n}^M(z, \alpha). \tag{110}$$

Counting the total number of terms is more complicated than for $g$. For a given $M$, the number of terms in $z^{2k+1}$ is the $(M-k-1)^{th}$ term of the expansion around the origin of the function $\frac{1+x^2}{(1-x)^2(1-x^3)}$, so after summation, the total number of terms is $F\left[\frac{(M+1)^3}{9} + \frac{3}{2}\right]$ where $F$ is the floor function.

We have easily identified a few of them, denoted by

$$J_{1,0}^M = \sum_{j=0}^{j_M} (-1)^j (j+1) \frac{z^{2j+1}}{\alpha^{2j}} \sum_{k=1}^{k_M} \left(\frac{4}{3\pi\alpha^3}\right)^k, \tag{111}$$

and other subterms which are very likely to belong to a more general one are the following:

$$-\frac{z}{\pi}\sum_{k\geq 0}(-1)^k 4\frac{k+2}{(2k+5)\alpha^{2k+5}},\tag{112}$$

$$\frac{z^3}{\pi}\sum_{k\geq 0}(-1)^k \frac{4}{3}\frac{(k+2)(k+3)}{\alpha^{2k+7}},\tag{113}$$

$$-\frac{z^5}{\pi}\sum_{k\geq 0}(-1)^k \frac{2}{15}\frac{(2k+7)(k+2)(k+3)(k+4)}{\alpha^{2k+9}}.\tag{114}$$

Further investigation is beyond the scope of this article.

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
