# Peer review of "Ground-state energy and excitation spectrum of the Lieb-Liniger model : accurate analytical results and conjectures about the exact solution"

_SciPost Physics, doi:SciPost Phys. 3, 003 (2017)_

## Round 2 · Referee Report · Anonymous · 2017-5-8

Strengths

(1) interesting results supported by a detailed analysis
(2) very good introduction to the problem

Weaknesses

(1) major one is a lack of a useful summary of the results

Report

Authors study integral equations describing properties of the ground state and excitations around the ground state of the Lieb-Liniger model. They use a series expansion, developed recently in this context by Z. Ristivojevic, to systematically find approximate solutions. They test the accuracy of the results by comparing them with numerical solutions. They evaluate a number of physically relevant quantities, like the ground state energy or local correlation functions, to high order in (the inverse of) the interaction parameter.

The presented results are interesting and the analysis is detailed. Their value is twofold. First as a step towards a better analytic understanding of the studied integral equations, especially the Love equation which has a long history in mathematical physics. Second, the authors provide explicit formulas for the aforementioned physical quantities. Most of these formulas were known in the strongly interacting limit and the main contribution of this work is to systematically extend their accuracy towards the weakly interacting regime.

The paper is well written and the results are worth communicating. There are few points that I would like the authors to address, these are listed below.

Requested changes

1) In section II.D authors introduce a notion of complexity of terms in the series expansion. I would like the authors to explain better whether (and if yes how) the complexity is connected with the form of the terms. Specifically, looking at eqs. (14) - (18) complexity seems to be related to the maximal power of k appearing in these equations. Is this correct?

2) There is a typo at the bottom of page 8: asuming -> assuming.

3) Below eq. (31) the authors write: "The fact that $g_2$ vanishes in the Tonks-Girardeau regime is once again a consequence of fermionization, as interactions induce a kind of Pauli principle and preclude that three bosons get very close from each other". This sentence is not very clear to me. $g_2$ describes probability of observing two bosons at the same position not three. I suppose the authors mean something else and would like to ask to clarify it.

3) This is a question of aesthetics but the end of section II.E.4 seems a bit too crumbled with inline math.

4) In the beginning of section III the authors write: "... we proceed with a more complicated and partially open problem, namely the characterisation of excitations above the ground state at zero temperature". I would like the authors to be more specific about what open problems do they mean.

5) Figures 8 and 10 would benefit from using different markers, and not only colour, to distinguish different types of discrete data points.

6) My main objection is the lack of a good summary of the results. In section IV the authors summarize they work but they do not point explicitly to main formulas. The formulas themselves are scattered through the text. In my opinion the quality of presentation would benefit from not only an explicit reference but also from repeating the results in the conclusion and thus providing a quick reference.

7) Finally, in the conclusions the authors write: "... the LeClair-Mussardo formalism can be deduced from the Bethe Ansatz, so that one does need to consider the sinh-Gordon model". I suppose there is a "not" missing between "does" and "need".

  • validity: high
  • significance: good
  • originality: good
  • clarity: high
  • formatting: excellent
  • grammar: excellent

Author:  Guillaume Lang  on 2017-05-30  [id 137]

(in reply to Report 2 on 2017-05-08)
Category:
answer to question

"1) In section II.D authors introduce a notion of complexity of terms in the series expansion. I would like the authors to explain better whether (and if yes how) the complexity is connected with the form of the terms. Specifically, looking at eqs. (14) - (18) complexity seems to be related to the maximal power of k appearing in these equations. Is this correct?"

-> The referee is right, the manuscript is elusive regarding this notion of complexity, because there is a priori no obvious definition thereof. In first approximation, it represents the level of difficulty to identify the patterns and coefficients of the different terms from scratch. Then, looking closer at the result, it seems that the complexity of first kind terms corresponds to the degree of the polynomial in k as pointed out by the referee. As far as the ground-state energy is concerned, complexity appears explicitly in the power of the denominator of Eq.(23). We have added some comments in the article regarding these points.

"2) There is a typo at the bottom of page 8: asuming -> assuming."

-> We have corrected the typo.

3) Below eq. (31) the authors write: "The fact that g2 vanishes in the Tonks-Girardeau regime is once again a consequence of fermionization, as interactions induce a kind of Pauli principle and preclude that three bosons get very close from each other". This sentence is not very clear to me. g2 describes probability of observing two bosons at the same position not three. I suppose the authors mean something else and would like to ask to clarify it.

-> The referee is right, actually this was a typo, instead of 'three' we meant 'two', and the sentence has been changed into: 'The fact that g_2 vanishes in the Tonks-Girardeau regime is once again a consequence of fermionization, as interactions induce a kind of Pauli principle and preclude that two bosons come in contact.'

"3) This is a question of aesthetics but the end of section II.E.4 seems a bit too crumbled with inline math."

-> Inline math has been transferred to equations for better visibility.

"4) In the beginning of section III the authors write: "... we proceed with a more complicated and partially open problem, namely the characterisation of excitations above the ground state at zero temperature". I would like the authors to be more specific about what open problems do they mean."

-> We have added 'analytical' before the word 'characterization', and added a sentence to explain that the aim is to compare the exact structure with the one predicted by field-theoretical techniques such as Luttinger liquid theory, to discuss their range of validity.

"5) Figures 8 and 10 would benefit from using different markers, and not only colour, to distinguish different types of discrete data points."

-> We have included triangular and square markers and modified their captions accordingly.

"6) My main objection is the lack of a good summary of the results. In section IV the authors summarize they work but they do not point explicitly to main formulas. The formulas themselves are scattered through the text. In my opinion the quality of presentation would benefit from not only an explicit reference but also from repeating the results in the conclusion and thus providing a quick reference."

-> We have added details and the main results to the summary as asked by the referee.

"7) Finally, in the conclusions the authors write: "... the LeClair-Mussardo formalism can be deduced from the Bethe Ansatz, so that one does need to consider the sinh-Gordon model". I suppose there is a "not" missing between "does" and "need"."

-> The referee is right, we have added the missing 'not'.

Attachment:

---

## Round 2 · Referee Report · Anonymous · 2017-5-8

Strengths

1) The paper is well written, and the results are presented clearly
2) The material is well contextualized, thanks to a large number of references and discussions on the existing literature
3) Many analytical results are presented in a unified way

Weaknesses

No weaknesses

Report

The authors consider several ground-state properties of the Lieb-Liniger model, focusing in particular on analytical perturbative expansions of known formulae.
While such expansions were already known in the literature, the authors extend them to significantly higher order, reaching in some cases remarkable accuracy in a wide range of the interaction parameter. Besides the per se interest, the results presented by the authors might be a useful starting point for computing perturbative expansions for complicated physical situations, beyond ground-state properties.

The paper is well written and the results clearly explained. Furthermore, I appreciate the effort put to provide an accurate set of references, which is extremely useful to contextualize the material in the manuscript. Thus, I recommend publication in SciPost Physics.

Requested changes

I have a few questions/remark listed below.

1) As the only suggested change, I would ask the authors to consider softening a few statements throughout the text. Indeed, in some parts of the manuscript, an inexperienced reader might be under the impression that the existing integral formulae describing the ground-state are difficult to solve numerically to good accuracy, while this is not the case. Indeed, simple iterative methods provides excellent numerical accuracy from large down to small interaction parameters, for almost all the ground-state properties.
For example, in the introduction one reads

“More problematically, a wide range of experimentally relevant, intermediate repulsive interactions is hardly accessed analytically by perturbation theory. This lack of accuracy affects the estimate of many physical quantities, such as the sound velocity or the Tan contact of the gas.”

I don’t agree with the implications of this sentence: in the ground-state almost all of the existing formulae could be evaluated numerically to high accuracy, even in the absence of analytical series expansions.

Of course, the fact that existing integral formulae could be numerically evaluated to high precision does not diminish the value of the results presented. The latter are a relevant contribution to the effort of pushing the analytical control as far as possible.

2) In section 2D, distinct groups of patterns of the series element are presented, as obtained from inspection of the general series. I think that the manuscript would benefit from a brief comment on how these distinct groups were identified. Was it only guess-work? Furthermore, I missed whether resummed expressions like (14) were proven to be generated from the general series (11), or only checked to be there for large values of M (and if this is the case, how large was M?).

3) After equations (22): could the authors be more clear on the definition of “complexity” used here?

4) As the authors suggest, their analytical methods could be used for analyzing other kinds of integral formulae, for example arising in study of thermal states. In this case, do they expect that the guess-work that I mentioned in the previous point will get more involved or could it be straightforwardly carried out?

  • validity: high
  • significance: high
  • originality: good
  • clarity: top
  • formatting: excellent
  • grammar: excellent

Author:  Guillaume Lang  on 2017-05-30  [id 138]

(in reply to Report 1 on 2017-05-08)
Category:
answer to question

"1) As the only suggested change, I would ask the authors to consider softening a few statements throughout the text. Indeed, in some parts of the manuscript, an inexperienced reader might be under the impression that the existing integral formulae describing the ground-state are difficult to solve numerically to good accuracy, while this is not the case. Indeed, simple iterative methods provides excellent numerical accuracy from large down to small interaction parameters, for almost all the ground-state properties.
For example, in the introduction one reads

“More problematically, a wide range of experimentally relevant, intermediate repulsive interactions is hardly accessed analytically by perturbation theory. This lack of accuracy affects the estimate of many physical quantities, such as the sound velocity or the Tan contact of the gas.”

I don’t agree with the implications of this sentence: in the ground-state almost all of the existing formulae could be evaluated numerically to high accuracy, even in the absence of analytical series expansions.

Of course, the fact that existing integral formulae could be numerically evaluated to high precision does not diminish the value of the results presented. The latter are a relevant contribution to the effort of pushing the analytical control as far as possible."

-> We have identified the statements that could be misleading and modified the text accordingly.

"2) In section 2D, distinct groups of patterns of the series element are presented, as obtained from inspection of the general series. I think that the manuscript would benefit from a brief comment on how these distinct groups were identified. Was it only guess-work? Furthermore, I missed whether resummed expressions like (14) were proven to be generated from the general series (11), or only checked to be there for large values of M (and if this is the case, how large was M?)."

-> Indeed our conjecture is a guess-work, based on the exact strong-coupling expansion to order 20 (that corresponds to M=9). Although there is no mathematical proof involved, we are pretty confident that the inferred expressions are exact, since they predict rational numbers for the expression of the energy that have up to 10 or more digits at numerator and denominator, coinciding with higher-order terms. We found no counter-example to our expressions and did not put any one that has not passed this 'blindfold test'.

"3) After equations (22): could the authors be more clear on the definition of “complexity” used here?"

-> The referee is right, the manuscript is elusive regarding this notion of complexity, because there is a priori no obvious definition thereof. In first approximation, it represents the level of difficulty to identify the patterns and coefficients of the different terms from scratch. Then, looking closer at the result, it seems that the complexity of first kind terms corresponds to the degree of the polynomial in k as pointed out by the other referee. As far as the ground-state energy is concerned, complexity appears explicitly in the power of the denominator of Eq.(23). We have added some comments in the article regarding these points.

"4) As the authors suggest, their analytical methods could be used for analyzing other kinds of integral formulae, for example arising in study of thermal states. In this case, do they expect that the guess-work that I mentioned in the previous point will get more involved or could it be straightforwardly carried out?"

-> As mentioned briefly in the text, at finite temperature the Lieb-Liniger equations become the Yang-Yang equations, that are coupled together in a non-trivial way. One can expect that the problem is considerably more involved then. It is possible that low-temperature expansions to order T^2 or more, following Sommerfeld's expansions, can be obtained, but the structure is assuredly more complex and this approach limited to small temperatures. This problem can thus be considered as a major open problem.

---

## Round 3 · Referee Report · Anonymous (Referee 3) · 2017-6-22

Strengths

1) The paper is well written, and the results are presented clearly
2) The material is well contextualized, thanks to a large number of references and discussions on the existing literature
3) Many analytical results are presented in a unified way

Weaknesses

No weaknesses

Report

The authors have addressed all of my comments and questions. I now recommend the manuscript for publication.

Requested changes

No requested changes

---

## Round 3 · Author Response

Dear Editors

we would like to resubmit a revised version of our article 'Ground-state energy and excitation spectrum of the Lieb-Liniger model : accurate analytical results and conjectures about the exact solution'. We have revised the paper following all the suggestions of the referees. A detailed answer to all the points raised by the referees follows.

Best regards Guillaume Lang, for the authors

Answer to the First Referee:

"1) In section II.D authors introduce a notion of complexity of terms in the series expansion. I would like the authors to explain better whether (and if yes how) the complexity is connected with the form of the terms. Specifically, looking at eqs. (14) - (18) complexity seems to be related to the maximal power of k appearing in these equations. Is this correct?"

-> The referee is right, the manuscript is elusive regarding this notion of complexity, because there is a priori no obvious definition thereof. In first approximation, it represents the level of difficulty to identify the patterns and coefficients of the different terms from scratch. Then, looking closer at the result, it seems that the complexity of first kind terms corresponds to the degree of the polynomial in k as pointed out by the referee. As far as the ground-state energy is concerned, complexity appears explicitly in the power of the denominator of Eq.(23). We have added some comments in the article regarding these points.

"2) There is a typo at the bottom of page 8: asuming -> assuming."

-> We have corrected the typo.

3) Below eq. (31) the authors write: "The fact that g2 vanishes in the Tonks-Girardeau regime is once again a consequence of fermionization, as interactions induce a kind of Pauli principle and preclude that three bosons get very close from each other". This sentence is not very clear to me. g2 describes probability of observing two bosons at the same position not three. I suppose the authors mean something else and would like to ask to clarify it.

-> The referee is right, actually this was a typo, instead of 'three' we meant 'two', and the sentence has been changed into: 'The fact that g_2 vanishes in the Tonks-Girardeau regime is once again a consequence of fermionization, as interactions induce a kind of Pauli principle and preclude that two bosons come in contact.'

"3) This is a question of aesthetics but the end of section II.E.4 seems a bit too crumbled with inline math."

-> Inline math has been transferred to equations for better visibility.

"4) In the beginning of section III the authors write: "... we proceed with a more complicated and partially open problem, namely the characterisation of excitations above the ground state at zero temperature". I would like the authors to be more specific about what open problems do they mean."

-> We have added 'analytical' before the word 'characterization', and added a sentence to explain that the aim is to compare the exact structure with the one predicted by field-theoretical techniques such as Luttinger liquid theory, to discuss their range of validity.

"5) Figures 8 and 10 would benefit from using different markers, and not only colour, to distinguish different types of discrete data points."

-> We have included triangular and square markers and modified their captions accordingly.

"6) My main objection is the lack of a good summary of the results. In section IV the authors summarize they work but they do not point explicitly to main formulas. The formulas themselves are scattered through the text. In my opinion the quality of presentation would benefit from not only an explicit reference but also from repeating the results in the conclusion and thus providing a quick reference."

-> We have added details and the main results to the summary as asked by the referee.

"7) Finally, in the conclusions the authors write: "... the LeClair-Mussardo formalism can be deduced from the Bethe Ansatz, so that one does need to consider the sinh-Gordon model". I suppose there is a "not" missing between "does" and "need"."

-> The referee is right, we have added the missing 'not'.

Answer to the Second Referee:

"1) As the only suggested change, I would ask the authors to consider softening a few statements throughout the text. Indeed, in some parts of the manuscript, an inexperienced reader might be under the impression that the existing integral formulae describing the ground-state are difficult to solve numerically to good accuracy, while this is not the case. Indeed, simple iterative methods provides excellent numerical accuracy from large down to small interaction parameters, for almost all the ground-state properties. For example, in the introduction one reads

“More problematically, a wide range of experimentally relevant, intermediate repulsive interactions is hardly accessed analytically by perturbation theory. This lack of accuracy affects the estimate of many physical quantities, such as the sound velocity or the Tan contact of the gas.”

I don’t agree with the implications of this sentence: in the ground-state almost all of the existing formulae could be evaluated numerically to high accuracy, even in the absence of analytical series expansions.

Of course, the fact that existing integral formulae could be numerically evaluated to high precision does not diminish the value of the results presented. The latter are a relevant contribution to the effort of pushing the analytical control as far as possible."

-> We have identified the statements that could be misleading and modified the text accordingly.

"2) In section 2D, distinct groups of patterns of the series element are presented, as obtained from inspection of the general series. I think that the manuscript would benefit from a brief comment on how these distinct groups were identified. Was it only guess-work? Furthermore, I missed whether resummed expressions like (14) were proven to be generated from the general series (11), or only checked to be there for large values of M (and if this is the case, how large was M?)."

-> Indeed our conjecture is a guess-work, based on the exact strong-coupling expansion to order 20 (that corresponds to M=9). Although there is no mathematical proof involved, we are pretty confident that the inferred expressions are exact, since they predict rational numbers for the expression of the energy that have up to 10 or more digits at numerator and denominator, coinciding with higher-order terms. We found no counter-example to our expressions and did not put any one that has not passed this 'blindfold test'.

"3) After equations (22): could the authors be more clear on the definition of “complexity” used here?"

-> The referee is right, the manuscript is elusive regarding this notion of complexity, because there is a priori no obvious definition thereof. In first approximation, it represents the level of difficulty to identify the patterns and coefficients of the different terms from scratch. Then, looking closer at the result, it seems that the complexity of first kind terms corresponds to the degree of the polynomial in k as pointed out by the other referee. As far as the ground-state energy is concerned, complexity appears explicitly in the power of the denominator of Eq.(23). We have added some comments in the article regarding these points.

"4) As the authors suggest, their analytical methods could be used for analyzing other kinds of integral formulae, for example arising in study of thermal states. In this case, do they expect that the guess-work that I mentioned in the previous point will get more involved or could it be straightforwardly carried out?"

-> As mentioned briefly in the text, at finite temperature the Lieb-Liniger equations become the Yang-Yang equations, that are coupled together in a non-trivial way. One can expect that the problem is considerably more involved then. It is possible that low-temperature expansions to order T^2 or more, following Sommerfeld's expansions, can be obtained, but the structure is assuredly more complex and this approach limited to small temperatures. This problem can thus be considered as a major open problem.

---

## Round 3 · List of Changes

- The conclusion has been modified to include a summary
- Typos noted by the referees have been corrected
- arXiv references published inbetween have been actuated

---

## Editorial Decision

published